# E3 ubiquitin ligase Bre1 couples sister chromatid cohesion establishment to DNA replication in *Saccharomyces cerevisiae*

Wei Zhang, Clarence Hue Lok Yeung, Liwen Wu, Karen Wing Yee Yuen*

School of Biological Sciences, The University of Hong Kong, Hong Kong, China

**Abstract** Bre1, a conserved E3 ubiquitin ligase in *Saccharomyces cerevisiae*, together with its interacting partner Lge1, are responsible for histone H2B monoubiquitination, which regulates transcription, DNA replication, and DNA damage response and repair, ensuring the structural integrity of the genome. Deletion of *BRE1* or *LGE1* also results in whole chromosome instability. We discovered a novel role for Bre1, Lge1 and H2Bub1 in chromosome segregation and sister chromatid cohesion. Bre1's function in G1 and S phases contributes to cohesion establishment, but it is not required for cohesion maintenance in G2 phase. Bre1 is dispensable for the loading of cohesin complex to chromatin in G1, but regulates the localization of replication factor Mcm10 and cohesion establishment factors Ctf4, Ctf18 and Eco1 to early replication origins in G1 and S phases, and promotes cohesin subunit Smc3 acetylation for cohesion stabilization. H2Bub1 epigenetically marks the origins, potentially signaling the coupling of DNA replication and cohesion establishment.

DOI: https://doi.org/10.7554/eLife.28231.001

*For correspondence:
kwyyuen@hku.hk

**Competing interests:** The authors declare that no competing interests exist.

## Introduction

Bre1 is a conserved E3 ubiquitin ligase containing a C3HC4 zinc-finger RING domain at its C-terminus, which forms a complex with Lge1 and associates with the E2 ubiquitin-conjugating enzyme Rad6 to mediate histone H2B monoubiquitination (H2Bub1) on lysine 123 (H2BK123) in *Saccharomyces cerevisiae* (*Hwang et al., 2003*; *Robzyk et al., 2000*; *Wood et al., 2003*). H2Bub1 is one of the histone posttranslational modifications that has been implicated in diverse cellular functions, including: transcription regulation (*Fleming et al., 2008*; *Minsky et al., 2008*; *Pavri et al., 2006*; *Sansó et al., 2012*) that is mediated through cycles of ubiquitination and deubiquitination (*Henry et al., 2003*; *Osley, 2006*) and by cross-talk effects on histone H3 methylation on residues K4 and K79 (*Briggs et al., 2002*; *Dover et al., 2002*; *Nakanishi et al., 2009*; *Ng et al., 2002*; *Sun and Allis, 2002*); DNA replication progression (*Trujillo and Osley, 2012*); modulation of nucleosome dynamics (*Chandrasekharan et al., 2009*; *Fierz et al., 2011*); DNA double-strand breaks (DSBs) repair (*Chernikova et al., 2010*; *Moyal et al., 2011*; *Nakamura et al., 2011*; *Northam and Trujillo, 2016*); DSB in meiosis (*Yamashita et al., 2004*); maintenance of functional, transcriptionally active centromeric chromatin in fission yeast (*Sadeghi et al., 2014*); methylation of kinetochore protein Dam1 (*Latham et al., 2011*); apoptosis (*Walter et al., 2010*); and cell size control (*Hwang et al., 2003*; *Jorgensen et al., 2002*).

The human homologs of yeast Bre1, the RING-finger proteins Rnf20 and Rnf40, form a heterodimer complex and are also required for H2Bub1 on lysine 120 (H2BK120) (*Zhu et al., 2005*). *RNF20* and *RNF40*, which are implicated as tumor suppressor genes, are mutated or misregulated in various types of cancers (*Johnsen, 2012*). H2Bub1 is also downregulated during tumor progression

**eLife digest** Most of the DNA in a cell is stored in structures called chromosomes. During every cell cycle, each cell needs to replicate its chromosomes, hold the two chromosome copies (also known as "sister chromatids") together before cell division, and distribute them equally to the two new cells. Each step must be executed accurately otherwise the new cells will have extra or missing chromosomes – a condition that is seen in many cancer cells and that can cause embryos to die. Since these processes are so essential to life, they are highly similar in a range of species, from single-celled organisms such as yeast to multicellular organisms like humans. However, it was not clear when and how sister chromatids first join together, or how this process is linked to DNA replication.

The DNA in the sister chromatids is wrapped around proteins called histones to form a structure known as chromatin. An enzyme called Bre1 plays roles in gene transcription and DNA replication and repair by adding ubiquitin molecules to a histone called H2B. Now, by using genetic, molecular and cell biological approaches to study baker and brewer yeast cells, Zhang et al. show that the activity of Bre1 helps to hold sister chromatids together. Specifically, Bre1 recruits proteins to the chromatin before and during DNA replication, which help to initiate replication and to establish cohesion between the sister chromatids. The ubiquitin molecule attached to H2B by Bre1 is also essential for establishing cohesion, acting as a mark that helps to link the two processes.

In the future it will be worthwhile to investigate whether genetic mutations that prevent sister chromatids adhering to each other is a major cause of the chromosome abnormalities seen in cancer cells. This knowledge may be useful for diagnosing cancers. Drugs that prevent the activity of Bre1 and other proteins involved in holding together sister chromatids could also be developed as potential cancer treatments that kill cancer cells by causing instability in their number of chromosomes.

DOI: https://doi.org/10.7554/eLife.28231.002

(*Thompson et al., 2013*), suggesting a role for H2Bub1 in tumor suppression. In budding yeast, *bre1∆* and *lge1∆* mutants have been identified in multiple genome-wide screens as exhibiting structural and numerical chromosomal instability (CIN) phenotypes (*Yuen et al., 2007*). The structural CIN phenotype involving gross chromosomal rearrangements (GCR) observed in *bre1∆* and *lge1∆* can be explained by the known functions of H2Bub1 in DNA damage response and repair, but the underlying cause of numerical CIN phenotypes involving whole chromosome gains or losses in *bre1∆* and *lge1∆* is currently not clear, though Bre1's function in replication origins has been implicated in mini-chromosome maintenance (*Rizzardi et al., 2012*). Accurate chromosome segregation requires the coordination of many cell-cycle-regulated processes, including sister chromatid cohesion, spindle assembly checkpoint, kinetochore function and centrosome function (*Yuen, 2010*).

*RNF20* was one of the five human homologs of yeast CIN genes that are somatically mutated in colorectal cancers (*Barber et al., 2008*). The other four genes regulate sister chromatid cohesion, affecting cohesin subunits *SMC1–SMC1L1*, *SMC3–CSPG6*, *SCC3–STAG3* and cohesin-loading complex subunit *SCC2–NIPBL*, implying that cohesion gene mutations are enriched in colorectal cancers (*Barber et al., 2008*). Whether *RNF20* also functions in sister chromatid cohesion is unknown.

Cohesion between the replicated sister chromatids is established from S phase until the onset of mitotic anaphase, which ensures that an identical set of genetic information is inherited by both daughter cells. Sister chromatid cohesion is mediated by a conserved multi-subunit ring-shaped protein complex called cohesin, which consists of four subunits: the coiled-coil proteins Smc1 and Smc3 are linked by the globular SMC hinge domains at one end, at the other end, the ATPase head domains bind to Scc1–Mcd1–Rad21–Klesin together with Scc3 (*Haering et al., 2002, 2004*; *Michaelis et al., 1997*; *Tóth et al., 1999*). Cohesin is proposed to hold DNA topologically (*Haering et al., 2008*). The cohesin complex is loaded onto chromosomes in late G1 by the cohesin-loading complex Scc2–Scc4 (*Ciosk et al., 2000*) through opening of the SMC hinge region (*Gruber et al., 2006*; *Nasmyth, 2011*). In budding yeast, cohesin preferentially accumulates between convergently transcribed genes and at centromeres (*Lengronne et al., 2004*; *Tanaka et al., 1999*). Establishment of sister chromatid cohesion during S phase requires an

essential acetyltransferase, Eco1/Ctf7, which acetylates the cohesin subunit Smc3 at K112 and K113 (*Rolef Ben-Shahar et al., 2008*; *Skibbens et al., 1999*; *Tanaka et al., 2000*; *Tóth et al., 1999*; *Unal et al., 2008*) to inhibit cohesin's interaction with the Wpl1–Pds5 complex, which destabilizes the cohesin on chromatin (*Rolef Ben-Shahar et al., 2008*; *Kueng et al., 2006*; *Rowland et al., 2009*; *Sutani et al., 2009*; *Terret et al., 2009*). In addition, two non-essential cohesion establishment pathways, including Ctf4 and Ctf18, contribute to cohesion establishment (*Hanna et al., 2001*; *Mayer et al., 2001*). Cohesion can no longer be established once replication is complete (*Uhlmann and Nasmyth, 1998*), except during DSBs in G2, when cohesin is recruited to DSBs for Eco1-dependent cohesion establishment and efficient break repair by homologous recombination (HR) (*Ogiwara et al., 2007*; *Ström et al., 2004*, *2007*; *Unal et al., 2007*). The destruction of cohesion at the onset of anaphase is mediated by separase-induced proteolysis of Scc1, thereby triggering the segregation of sister chromatids (*Nasmyth and Haering, 2009*; *Peters et al., 2008*).

Emerging evidence suggests that establishment of cohesion between sister chromatids is coupled to replication fork progression. A number of replication proteins, including the replication factor C (RFC) core subunit Rfc4 (*Mayer et al., 2001*), the DNA sliding clamp Proliferating Cell Nuclear Antigen (PCNA) (encoded by *POL30*) (*Lengronne et al., 2006*; *Moldovan et al., 2006*), the helicase Chl1 involved in processing Okazaki fragments (*Samora et al., 2016*; *Skibbens, 2004*), the leading-strand DNA polymerase ε (*Edwards et al., 2003*), the replication checkpoint proteins Tof1 and Csm3 (*Mayer et al., 2004*), and subunits of the origin recognition complex (ORC) subunits Orc2 and Orc5 (*Shimada and Gasser, 2007*; *Suter et al., 2004*) play important roles in sister chromatid cohesion.

In turn, cohesion establishment factors localize to replication forks, affecting fork progression and stability (*Gambus et al., 2009*; *Lengronne et al., 2006*; *Terret et al., 2009*). Smc3 acetyltransferase Eco1 associates with the replication fork through PCNA (*Moldovan et al., 2006*; *Skibbens et al., 1999*). Ctf18, a component of the replication factor C (RFC$^{Ctf18}$) complex, can load and unload PCNA (*Bylund and Burgers, 2005*; *Lengronne et al., 2006*; *Mayer et al., 2001*; *Murakami et al., 2010*; *Shiomi et al., 2007*; *Terret et al., 2009*) and physically interacts with Eco1 (*Kenna and Skibbens, 2003*). The localization of Ctf18 at replication origins partially depends on Ctf4 (*Lengronne et al., 2006*). Ctf4 is a component of the replisome progression complex (RPC) (*Gambus et al., 2006*) that recruits DNA polymerase α (Polα)/primase for lagging-strand synthesis (*Gambus et al., 2009*; *Zhu et al., 2007*) and recruits Chl1 helicase (*Samora et al., 2016*) to the replisome through its physical association with the GINS (go-ichi-ni-san) complex. This complex is part of the Cdc45–Mcm2-7–GINS (CMG) helicase complex that is important for origin unwinding, establishment of the replication fork at origins and fork progression (*Gambus et al., 2009*). Ctf4 in turn depends on GINS and the replication factor Mcm10 for its localization (*Perez-Arnaiz et al., 2016*; *Terret et al., 2009*; *Wang et al., 2010*; *Zhu et al., 2007*). Mcm10's localization to origins is facilitated by the presence of inactive Mcm2-7 complex (*Douglas and Diffley, 2016*; *Ricke and Bielinsky, 2004*; *Wohlschlegel et al., 2002*). Next, Mcm10 recruits Cdc45 and GINS to inactive Mcm2-7 complex, and activates the CMG replicative helicase (*Perez-Arnaiz et al., 2016*; *Quan et al., 2015*; *Thu and Bielinsky, 2014*). Thus, Mcm10 is crucial in replication initiation and elongation.

Previous work has shown that H2Bub1 is not required for the association of pre-replication complex (ORC and Mcm4) and Cdc45 with origins in G1 phase, but is required for Mcm4, Cdc45, Psf2 (a component of GINS), Polα, Polε, RPA and Spt16's chromatin association in S phase, both for replisome stability and for nucleosome assembly onto nascent DNA at active replication forks (*Trujillo and Osley, 2012*). Whether Bre1 and H2Bub1 could affect sister chromatid cohesion through its function in DNA replication has not been explored.

Here we show that Bre1 RING-domain- and Lge1-mediated H2Bub1 is critical for accurate chromosome segregation, and specifically sister chromatid cohesion. Bre1's role in G1 and S phase contributes to cohesion establishment, but it is dispensable for cohesin component loading. Bre1 facilitates the localization of the upstream replication factor Mcm10 and cohesion establishment factors (Ctf18, Ctf4 and Eco1) to chromatin and early replication origins in G1 and S phases. The recruitment of these factors by Bre1 not only stabilizes the replisome progression complex, advancing the replication fork in S phase (*Trujillo and Osley, 2012*), but also couples the establishment of sister cohesion to maintain whole-chromosome stability.

## Results

### The E3 ubiquitin ligase Bre1–Lge1 complex is required for accurate chromosome segregation and sister chromatid cohesion

The genome-wide CIN screens in budding yeast have revealed that the E3 ubiquitin ligase *BRE1* and its interacting partner *LGE1* are important for maintaining chromosome stability (*Yuen et al., 2007*). To assess the chromosome transmission fidelity (CTF) of *bre1Δ* and *lge1Δ*, we monitored artificial chromosome fragment loss rate as described previously (*Spencer et al., 1990*; *Yuen et al., 2007*). Haploid cells in *ade2–101ochre* mutation background are red in color. An artificial chromosome III fragment (CF) with a yeast centromere and telomeres at the ends, resembling natural chromosomes' structure and stability, was introduced to the cells. The CF also contains a selectable marker and the *SUP11* tRNA suppressor gene, which suppresses the *ochre* mutation (*Spencer et al., 1990*). Cells containing the CF are white, whereas cells that have lost the CF are red. Thus, red sectors within a white colony indicate that the CF is lost in some mitoses during the formation of the colony. To quantify the CF loss rate per cell division, individual cells from selective medium were plated onto non-selective medium, and the percentage of colonies that were half-red or more than half-red on non-selective medium, representing cells that have lost the CF during the first cell division, was calculated (*Figure 1A*). Consistent with prior work (*Yuen et al., 2007*), deletion of *BRE1* or *LGE1* resulted in significant CF loss rates (0.78% and 0.85%, respectively) (*Figure 1A*), which were 5.7- and 6.3-fold higher than that in wildtype cells (0.13%), suggesting a role for Bre1–Lge1 in accurate chromosome segregation.

To dissect whether chromosome loss arises from a cohesion defect in *BRE1-* or *LGE1*-deleted cells, we utilized a *MATa* haploid strain containing Lac operator tandem repeats integrated 22 kb from the centromere of chromosome III and expressing a GFP–Lac repressor fusion protein as described *by Straight et al. (1996)*. The separation of the two sister chromatids can be visualized by the GFP signals during G2/M phase. If sister chromatid cohesion is normal, only one GFP focus can be observed due to the tight tethering of replicated sister chromatids by cohesion. However, two GFP foci can be seen if the sister chromatids prematurely separate (*Figure 1B*). Only 3.5% of WT cells with a large bud (60–75 min release from G1 arrest by alpha-factor [α-F]) had two GFP signals, and fluorescence-activated cell sorter (FACS) analysis showed that these large budded cells have replicated DNA content (*Figure 1C*). By contrast, *bre1Δ* and *lge1Δ* showed significant increases in the frequency of G2/M cells containing two GFP signals (20.2% and 19.9%, respectively) (*Figure 1D*).

Alpha-factor (α-F) was added back at 60 min post G1 release to arrest cells in the next G1 phase, and mitotic anaphase and cytokinesis finished by 120–150 min after G1 release (*Figure 1C*). Each G1 cell should contain one GFP focus if chromosomes are accurately segregated, whereas cells with mis-segregated chromosome III may have no GFP focus or two GFP foci (*Figure 1B*). In WT cells, 99% of cells contained only one GFP dot, indicative of proper chromosome separation. By contrast,~9% of *bre1Δ* or *lge1Δ* cells had two or no GFP signals, indicating a gain or loss of chromosome (*Figure 1E*). Together, these results demonstrate that the chromosome missegregation phenotype exhibited by *BRE1* or *LGE1* deletion could be caused by the defect in sister chromatid cohesion, though we cannot rule out the possibility that other cellular functions of Bre1 could also contribute to whole-chromosome stability.

### Bre1's role in G1 and S phase is important for sister chromatid cohesion

The cohesin ring complex is loaded onto chromosomes in late G1, whereas sister chromatid cohesion is established in S phase, and maintained through G2/M before anaphase (*Mehta et al., 2012*). To determine the cell-cycle stages at which Bre1 is important for sister chromatid cohesion, we exploited the Auxin-Inducible Degron (domain for inducing degradation) (AID) system (*Nishimura et al., 2009*) to conditionally control the expression of Bre1 at specific cell-cycle stages and examined the cohesion phenotype in G2/M-arrested cells. We constructed an AID*−9Myc tag at the C-terminus of Bre1 at the endogenous locus and expressed F-box Transport Inhibitor Response 1 (TIR1) from rice *Oryza sativa* (OsTIR1) in a *MATa* haploid strain containing the GFP–LacI: LacO system for cohesion assay. Auxin binds to AID* and OsTIR1, which interacts with the E3 ubiquitin ligase SCF (Skp1, Cullin and F-box) and targets AID*−9Myc-tagged Bre1 for polyubiquitination

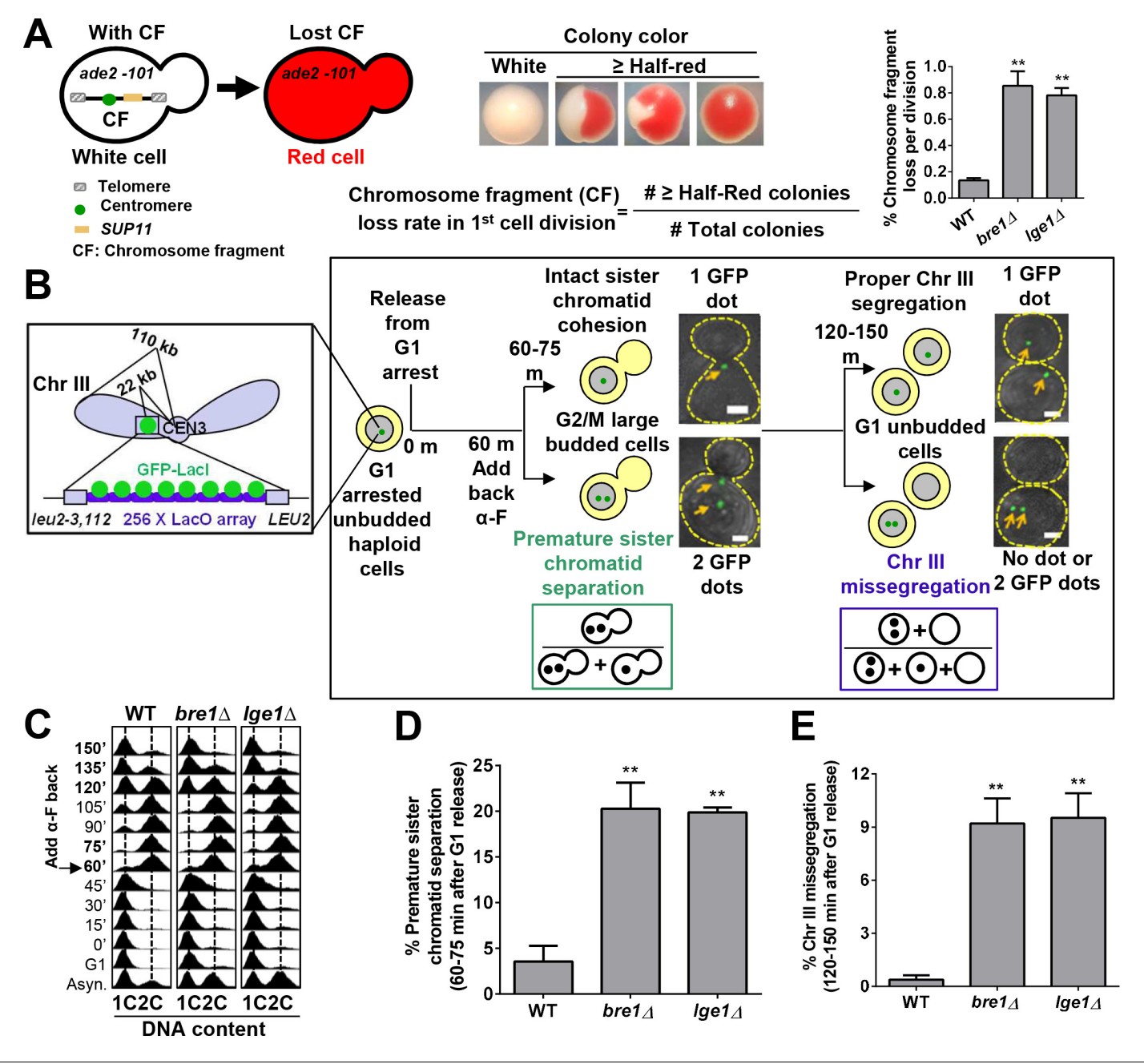

**Figure 1.** The E3 ubiquitin ligase Bre1 and its interacting protein Lge1 are required for accurate chromosome segregation and sister chromatid cohesion. (A) Chromosome transmission fidelity (CTF) phenotype was evaluated by the colony color-sectoring assay. Chromosome fragment (CF) loss rates in the first cell division in wild-type (WT), *bre1Δ* and *lge1Δ* were quantified by the number of half-red and more than half-red colonies divided by the total number of colonies. At least 2000 cells were scored in each experiment. The data shown represent the average of three independent experiments. Error bars, standard errors of the mean (SEM). Statistical significance was calculated by Student's t-test. Significant differences with WT are indicated by asterisks (**p<0.01). (B) Schematic diagram and flowchart of the sister chromatid cohesion assay and chromosome segregation assay examining GFP–Lac repressor (GFP–LacI) binding to Lac operator (LacO) integrated at 22 kb from centromere 3 (CEN3). *MATa* haploid yeast cells containing LacO and GFP–LacI in early log phase were arrested in G1 with alpha factor (α-F) and then released into YPD medium. α-F was added back at 60 min after release from G1 arrest to re-arrest cells at G1 phase in the next cell cycle. Samples were collected every 15 min for fluorescence-activated cell-sorting (FACS) analysis of DNA content and GFP fluorescence imaging. Representative fluorescent and bright field images were superimposed. Scale bar, 1 µm. Cells at 60–75 min after release from G1 arrest had large buds, and FACS showed that they were at G2/M phase. Sister chromatid cohesion was assessed. If sister chromatids had cohesion, only one GFP focus was observed. If they prematurely separated, two GFP foci were observed. The frequency of sister chromatid premature separation was calculated by the number of G2/M cells containing two GFP foci

*Figure 1 continued on next page*

Figure 1 continued

divided by the total number of G2/M cells. Most of the cells at 120–150 min after release from G1 arrest had no bud, suggesting that they had completed cytokinesis. They were assessed for chromosome segregation. Cells with more or less than one GFP focus had missegregated Chromosome III (ChrIII). The ratio of ChrIII missegregation was calculated by the number of G1 cells containing no GFP focus and two GFP foci divided by the total number of unbudded G1 cells. (C) Cell cycle progression of WT, *bre1Δ* and *lge1Δ* used in *Figure 1B*. Cells were arrested in G1 by alpha-factor (α-F) for 3 hr. Cells were washed and released into YPD medium. α-F was added back to the culture at 60 min post G1 release to restrict cells at G1 in the next cell cycle. Samples were collected every 15 min, and stained with propidium iodide for FACS analyses. (D) Frequency of sister chromatid premature separation in large budded cells at 60–75 min after release from G1 arrest in WT, *bre1Δ* and *lge1Δ* strains. At least 100 cells were scored for each sample. The data shown represent the average of three independent experiments. Error bars, SEM. Statistical analysis was performed using Student's t-test. Asterisks indicate significant differences from WT (**p<0.01). (E) Frequency of ChrIII missegregation in unbudded cells at 120–150 min after release from G1 arrest in WT, *bre1Δ* and *lge1Δ* strains. At least 100 cells were scored for each sample. The data shown represent the average of three independent experiments. Error bars, SEM. Statistical analysis was performed using Student's t-test. Asterisks indicate significant differences from WT (**p<0.01).

DOI: https://doi.org/10.7554/eLife.28231.003

and proteasome-mediated degradation. AID*−9Myc-tagged Bre1 and OsTIR expression did not affect growth and sister chromatid cohesion function (*Figure 2A and B*), but affected G1-S transition cyclin gene expression mildly (*Figure 2—figure supplement 1A*) (*Zimmermann et al., 2011*) and hydroxyurea (HU) sensitivity (*Figure 2—figure supplement 1B*). To assess the efficiency of auxin-induced Bre1-AID*−9Myc degradation, we monitored the degradation time course after treatment with 1 mM auxin in combination with G1, S and G2/M arrest. First, Bre1-AID*−9Myc-expressing cells were synchronized in early G1 phase by adding α-F for 3 hr, and then released into media containing α-F or HU for G1 or S phase arrest, respectively, together with 1 mM auxin to induce Bre1 degradation. Alternatively, to induce Bre1 degradation in G2/M phase, G1-arrested cells were released into HU-containing medium for 2 hr, and then released into Nocodazole (Noc)-containing medium with the addition of 1 mM auxin. Samples were collected every 15 min for FACS analysis of DNA content (*Figure 2C*) and for western blotting analysis of Bre1 protein level (*Figure 2D*). Specifically, over 90% of Bre1 was degraded after 45 min in G1, 75 min in S and 60 min in G2/M phase after auxin induction (*Figure 2D*).

To examine the timing of Bre1's function in the cohesion cycle, we degraded Bre1 at specific cell-cycle stages. Once the arrest in G1, S or G2/M was achieved by α-F, HU or Noc, respectively, with or without auxin induction, the medium was washed and released into the next cell cycle. Samples were collected every 30 min for FACS analysis of DNA content (*Figure 2E*) and western blotting analysis of Bre1 protein level to confirm the recovery of Bre1 (*Figure 2F*). Finally, cohesion phenotype in G2/M-arrested cells was assessed to determine whether Bre1 is functional in the G1, S, or G2/M phase. The no degradation control (without auxin in all stages) resulted in only 4.6% of cells showing a cohesion defect (*Figure 2G*), which was comparable to rates in WT cells (*Figure 1D*, Figure 5B and C), suggesting that no cohesion defect was caused by the effects of cell cycle arrest by different drugs. As expected, degradation of Bre1 in all cell-cycle stages showed 19.2% of cells with a cohesion defect, which was similar to the proportion in *bre1Δ*. Degradation of Bre1 in G1 only or in G1 and S phases showed similar proportions of cohesion defects (20.7% and 17.1%, respectively), which were comparable to the proportion of cells in which Bre1 was degraded in all cell-cycle stages (19.2%), suggesting that Bre1's role in G1 phase contributes to sister chromatid cohesion. To our surprise, degradation of Bre1 in S phase only or in S and G2/M phases resulted in fewer cells with cohesion defects (12.4% and 10.2%, respectively), but still significantly more defective cells compared to control. The occurrence rate of the cohesion defect caused by degradation of Bre1 in G1 and S phases (17.1%) was significantly higher than that in cells in which Bre1 is degraded in S only (12.4%). It is possible that for degradation of Bre1 in S phase, the residual level of Bre1 during early S-phase time points may suffice for some function (*Figure 2F*). However, degradation of Bre1 in G2/M phase only resulted in an occurrence rate for cohesion defects similar to that in the the negative control, implying that Bre1 is not required in G2/M phase for cohesion. Taken together, these findings demonstrated that Bre1's role in sister chromatid cohesion is most prominent in G1 phase, but also in S phase, consistent with the timing of cohesin loading in G1 phase or cohesion establishment in S phase, but Bre1 is not required in G2/M phase for cohesion maintenance. To further delineate the role of Bre1 in cohesion, we tested its effects on cohesin loading and cohesion establishment.

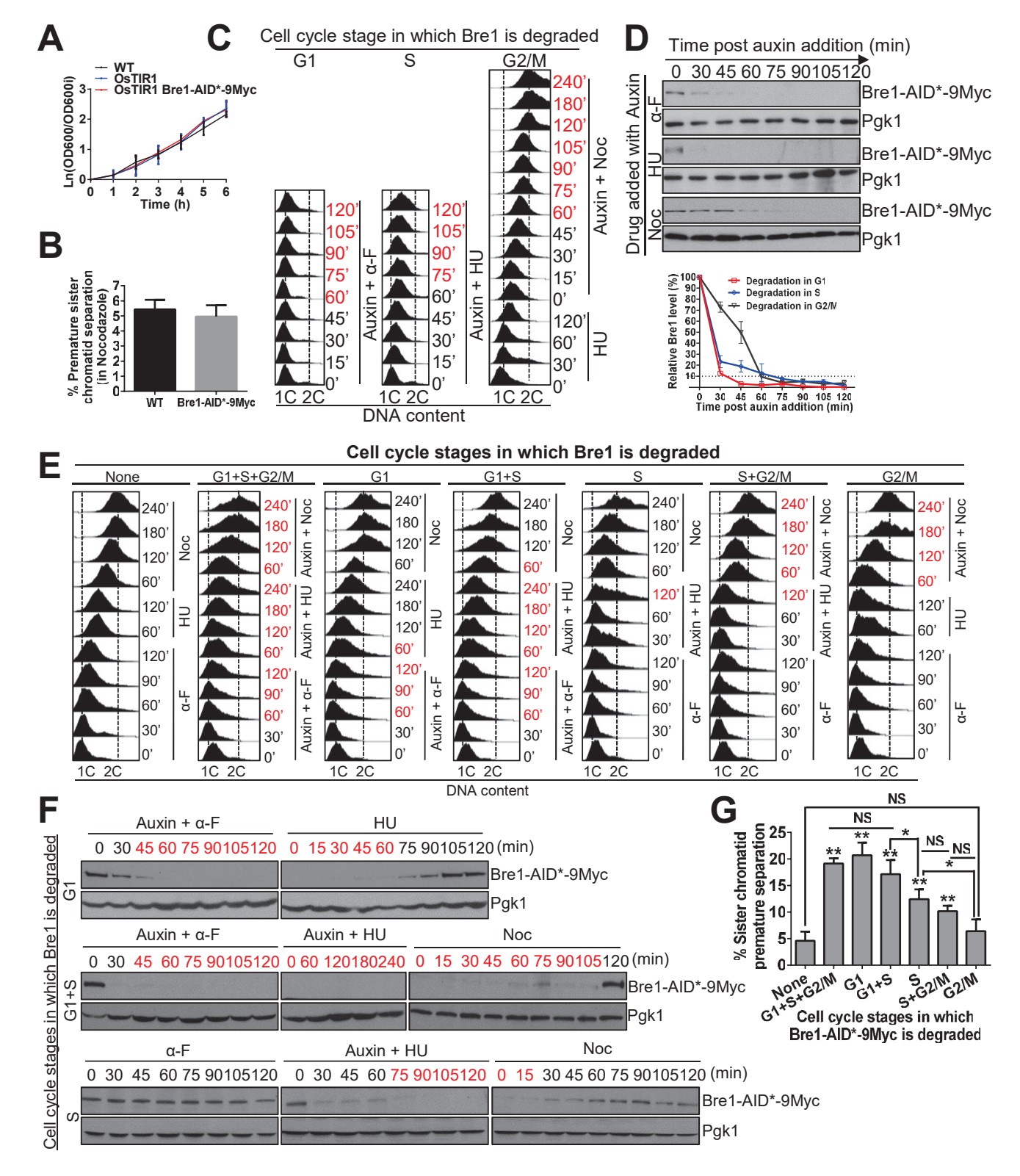

**Figure 2.** The function of Bre1 in G1 and S phase is important for sister chromatid cohesion. (**A**) Growth curve for the untagged wild-type (WT), OsTIR1 and OsTIR1 Bre1-AID*−9Myc strains. Early log phase cells were diluted to OD600 = 0.1. The OD600 was measured every 1 hr for 6 hr. The ln of the OD600/OD600i (OD600i = OD600 at '0' h) was calculated and plotted in the graph. (**B**) Frequency of sister chromatid premature separation in G2/M phase in WT and Bre1-AID*−9Myc cells. At least 100 cells were scored for each sample. Error bar represents SEM from three independent experiments.
*Figure 2 continued on next page*

*Figure 2 continued*

Statistical significance was analyzed by Student's t-test between WT and Bre1-AID*−9Myc. (C) Depletion of Bre1 in G1-, S- and G2/M-arrested cells through auxin-dependent degradation. Cells containing Bre1-AID*−9Myc were first arrested in G1 with α-F for 3 hr and washed with water and release. Next, to induce Bre1 degradation in G1 or S phase, 1 mM auxin was added to medium containing α-F or HU for 2 hr, respectively. To induce Bre1 degradation in G2/M phase, cells from G1 arrest was released into HU-containing medium for 2 hr, and then released into Noc-containing media with the addition of 1 mM auxin for 4 hr. Samples were collected every 15 min for FACS analyses and western blotting analyses (D). The time point at which > 90% Bre1 protein is degraded (from [D]) is highlighted in red. (D) Western blotting analysis of the protein level of Bre1-AID*−9Myc after auxin induction in different cell-cycle-arrested stages using anti-Myc antibody. Pgk1 served as the loading control. Quantitative analysis of the relative levels of Bre1-AID*−9Myc levels over time (normalized to loading control) at the indicated time points (100% at time 0 min post auxin addition) from three independent experiments were analyzed using Image J software and plotted. Error bars represent SEM of the mean. The dashed line indicates when 90% of the Bre1 protein is degraded. (E) FACS analysis of DNA contents at the indicated time points in the cell-cycle-specific auxin-induced Bre1 degradation and cohesion experiment in (G). According to (D), the time points after the initiation of auxin treatment at which > 90% of the Bre1 protein is degraded, and according to (F), the samples that are in recovery (>90% Bre1 protein is degraded) are highlighted in red. (F) Degradation and recovery of Bre1-AID*−9Myc in which Bre1 is degraded in G1 only, in G1 and S, or in S only. Western blotting analyses of the protein level of Bre1-AID*−9Myc after auxin induction in different cell cycle-arrested stages, and after auxin removal, using anti-Myc antibody. Pgk1 served as the loading control. The time points in degradation or recovery at which > 90% Bre1 protein is degraded are highlighted in red. (G) Frequency of sister chromatid premature separation in G2/M phase after Bre1 is degraded during the indicated cell-cycle stages. At least 100 cells were scored for each sample. Error bars represents SEM from three independent experiments. Statistical significance was analyzed by Student's t-test between no degradation control and degradation at different stages, or between each of the indicated degradation stages. **p<0.01; * p<0.05; NS, no significant difference among the indicated degradation stages.

DOI: https://doi.org/10.7554/eLife.28231.004
The following figure supplement is available for figure 2:

**Figure supplement 1.** Characterization of Bre1-AID*−9Myc function.
DOI: https://doi.org/10.7554/eLife.28231.005

## Bre1 is dispensable for cohesin loading

Cohesin associates with chromatin in late G1, then accumulates at regions of convergent transcription (*Lengronne et al., 2004*). Bre1 regulates the transcription of genes involved in G1-S transition (*Zimmermann et al., 2011*), but not that of cohesin components or coehsion establishment genes. The change in the transcription pattern of the G1-S transition genes may affect the binding of cohesin on chromatin. Next, we constructed strains expressing Scc1 or Smc3 3HA-tagged and examined their chromatin enrichment in G1-, S- and G2/M-arrested cells in WT and *bre1Δ* by chromatin spreads. The protein levels of Scc1-3HA and Smc3-3HA in *bre1Δ* were comparable to those in WT cells (*Figure 3—figure supplement 1B and D*), consistent with the mRNA level result (*Figure 3—figure supplement 1A*). As expected and consistent with the study by *Baetz et al. (2004)*, there was no detectable Scc1-3HA and Smc3-3HA signal in α-F-arrested early G1 phase WT or *bre1Δ* cells. The cohesin components were associated with chromatin in S and G2/M-arrested cells in WT cells (*Prinz et al., 1998*), which were unaffected by the deletion of *BRE1* (*Figure 3—figure supplement 1C,E,F and G*).

As cohesin subunit Scc1 is transiently enriched at active early replication origins and spreads along DNA as replication fork progresses (*Tittel-Elmer et al., 2012*), and as Bre1 is also present at origins of replication throughout the cell cycle (*Trujillo and Osley, 2012*), we tested whether Bre1 affects the enrichment of Scc1 at early origins in HU-arrested S phase cells using chromatin immunoprecipitation followed by quantitative real-time PCR (ChIP-qPCR). In agreement with the chromatin spreads result, the enrichment of Scc1–3HA at early origins (ARS305 and 306) was not affected by *BRE1* deletion (*Figure 3—figure supplement 1H*). Therefore, our results suggest that Bre1, like cohesion establishment factor Eco1/Ctf7 (*Tóth et al., 1999*), is dispensable for cohesin binding to chromatin.

To confirm the physical interaction of Bre1 with Smc3 in a yeast two-hybrid experiment (*Newman et al., 2000*), we performed co-immunoprecipitation in a strain containing Flag–Bre1 and Smc3–13Myc using anti-Myc antibody to immunoprecipitate Smc3–13Myc. We cannot, however, confirm interaction between Bre1 and Smc3 (*Figure 3—figure supplement 1I*).

# Bre1 affects the recruitment of cohesion establishment factors Ctf4, Ctf18 and Eco1 to early replication origins and promotes the acetylation of Smc3

Our Bre1 degron mutant experiment suggested that Bre1 is important in G1 and S phases, and may be involved in sister chromatid cohesion establishment in S phase. To elucidate whether Bre1 affects the recruitment of cohesion establishment factors Ctf4, Ctf18 and Eco1 to chromatin, chromatin spread was analyzed at different arrested cell cycle stages, and we found that Ctf4, Ctf18 and Eco1 associate with chromatin during all stages of the cell cycle in WT cells (*Figure 3A–C* and *Figure 3—figure supplement 2A–F*). This observation is consistent with previous studies (*Hanna et al., 2001; Tóth et al., 1999*), but our chromatin spread assay did not detect the degradation of Eco1 in G2/M phase as shown in a previous study (*Lyons and Morgan, 2011*). Nevertheless, the association of Ctf4 and Ctf18 with chromatin in S phase was significantly reduced in *bre1Δ* cells, whereas this association was not affected in G1 and G2/M phases (*Figure 3A–C*). The association of Eco1 with chromatin was reduced in *bre1Δ* cells in both G1 and S phases. As Ctf4, Ctf18 and Eco1 also localize at early replication origins and at forks in HU-treated early S-phase cells (*Lengronne et al., 2006*), and as Bre1 and H2Bub1 are present at origins (*Trujillo and Osley, 2012*), we hypothesized that Bre1 is required for the recruitment of Ctf4, Ctf18 and Eco1 to early origins. Their occupancy at origins was measured by ChIP in HU-arrested cells. The occupancy of Ctf4, Ctf18 and Eco1 at early origins ARS305 and ARS306, and of Eco1 at early origin flanking regions was reduced significantly in *bre1Δ* compared to WT (*Figure 3D–G*). In addition, the occupancy of Eco1 at early origin was also reduced significantly in G1 cells (*Figure 3H* and *Figure 3—figure supplement 2G*). Ctf4, Ctf18 and Eco1 were present at lower, but comparable, levels at a late origin ARS501 in *WT* and *bre1Δ* cells. The decreased occupancy of cohesion establishment factors at chromatin or early origin is not due to any change in mRNA or protein levels (*Figure 3—figure supplement 1A* and *Figure 3—figure supplement 2A–C*). Our results suggest that Bre1 facilitates the localization of these cohesion establishment factors to the chromatin, and specifically to the early origins. In the absence of Bre1, however, a proportion of cohesion factors remained on chromatin and at origins, suggesting that there are probably redundant pathways that function in their recruitment.

Since Ctf18 and Eco1 are both required for the acetylation of Smc3 during S phase to generate a stably chromosome-bound cohesin pool for enduring sister chromatid cohesion (*Beckouët et al., 2010; Terret et al., 2009*), we postulated that Bre1 may also affect Smc3 acetylation in S phase. We monitored acetylated Smc3 in WT and *bre1Δ* cells by western blotting using anti-Smc3-Ac antibody. In the absence of Bre1, Smc3 acetylation was significantly diminished, whereas the protein levels of Smc3-13Myc in WT and *bre1Δ* cells were similar (*Figure 3I*). These results suggest that Bre1 affects the acetylation of Smc3, but not the protein level of Smc3 or its association with chromatin. Collectively, we showed that despite the fact that Bre1 is dispensable for cohesin association to chromatin in S phase, it is important for recruiting cohesion establishment factors to the replication origins in G1 and S phases, and for Smc3 acetylation.

However, if Bre1's only function in cohesion is to facilitate Smc3 acetylation, removal of *wpl1Δ*, the destabilizer of chromatin-bound cohesin, may rescue *bre1Δ*'s cohesion defect, as for *ctf18Δ* (*Borges et al., 2013*). Surprisingly, *wpl1Δ* partially rescues *bre1Δ*'s cohesion defect at 60–75 min after G1 release, suggesting that Bre1 could play a role in reducing cohesion turnover on chromatin, counteracting Wpl1 (*Figure 3—figure supplement 2H*). Alternatively, it may reflect a slight delay in the cell cycle progression of *wpl1Δ bre1Δ* (*Figure 3—figure supplement 2I*). However, *wpl1Δ* has 9.2% of cells with a cohesion defect at 90 min after G1 release, whereas *wpl1Δ bre1Δ* has 18.8%, similar to the proportion in *bre1Δ* alone. This suggests that stabilizing cohesin alone on chromatin does not fully rescue the cohesion defect in *bre1Δ* as it does in *ctf4Δ* (*Borges et al., 2013*).

To investigate the genetic relationship between *BRE1* and the non-essential cohesion establishment factors *CTF4* and *CTF18* (*Hanna et al., 2001; Mayer et al., 2001; Xu et al., 2007*), we monitored the cohesion phenotypes in single and pair-wised double mutants. The cohesion defect in *bre1Δ* or *lge1Δ* was slightly less severe than that in *ctf4Δ* (23.7%) or *ctf18Δ* (28.7%) (*Figure 3J*). The cohesion phenotype difference in single mutants is consistent with the chromatin association results, suggesting that other parallel pathways independent of *bre1Δ* can help to recruit Ctf4 and Ctf18. Interestingly, cohesion defects in the double mutants, *ctf18Δ bre1Δ* (23.5% of cells), *ctf4Δ lge1Δ* (19.9%) and *ctf4Δ bre1Δ* (23.9%), were not significantly more frequent than those in the more severe

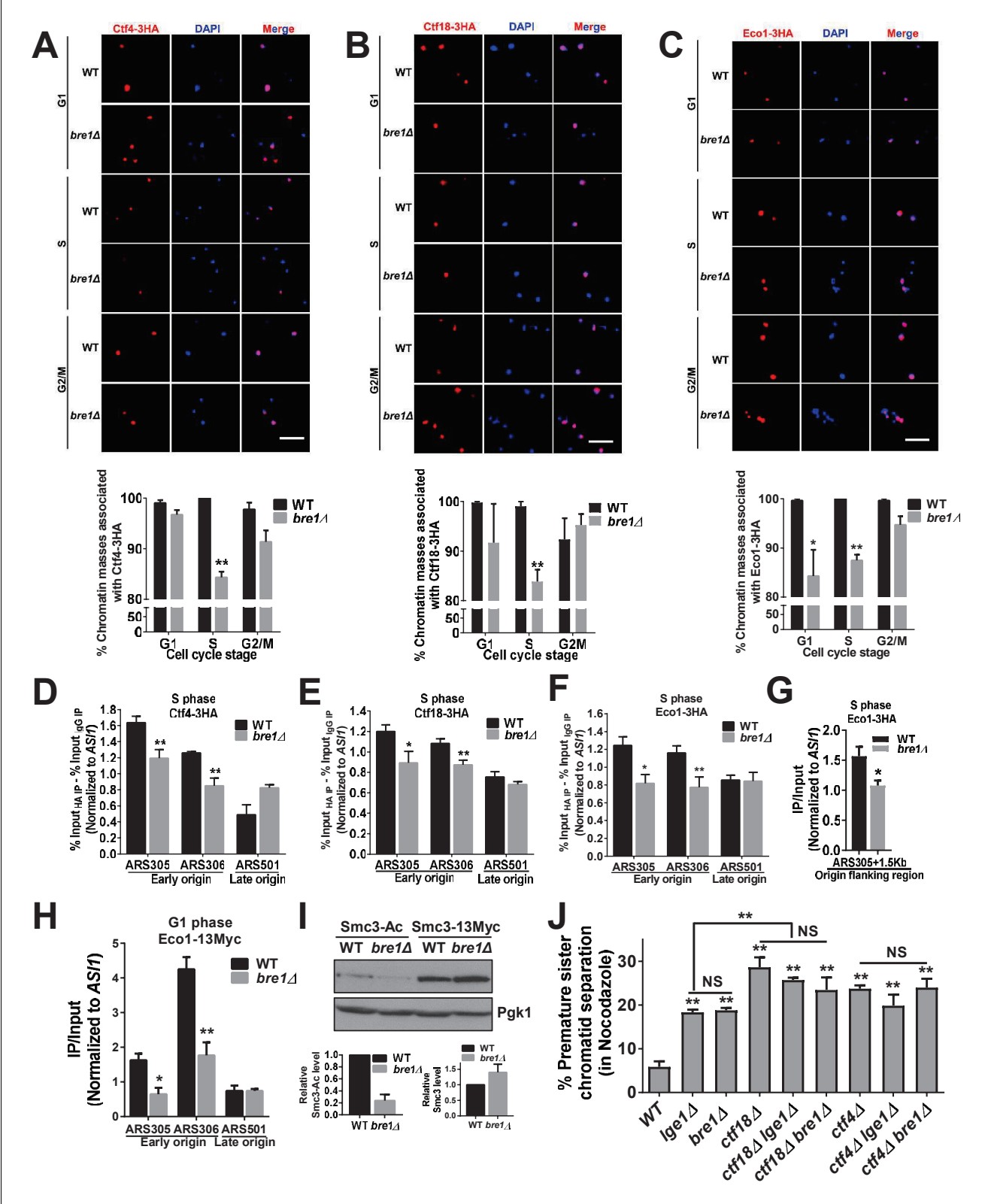

**Figure 3.** Bre1 recruits cohesion establishment factors Ctf4, Ctf18 and Eco1 to chromatin and replication early origins, and promotes the acetylation of Smc3. (A–C) Chromatin spreads analysis of the association of Ctf4-3HA (A), Ctf18-3HA (B) and Eco1-3HA (C) with chromatin in G1-, S- and G2/M-arrested cells. The frequencies of chromatin-associated Ctf4-3HA, Ctf18-3HA and Eco1-3HA were quantified. At least 100 chromatin masses were scored. Error bars represent the SEM from three independent experiments. * and ** indicate p<0.05 and p<0.01, respectively, as determined by

*Figure 3 continued on next page*

*Figure 3 continued*

Student's t-test. Scale bar, 1 µm. (D–G) Chromatin immunoprecipitation analysis of the occupancy of Ctf18-3HA (D), Ctf4-3HA (E) and Eco1-3HA (F and G) in wild-type (WT) and *bre1Δ* cells at replication early origins ARS305 and ARS306, late origin ARS501 (D–F) and origin flanking region ARS305 +1.5 kb (G) after arrest of cells in HU for 3 hr. Immunoprecipitation (IP) signals at ARS sequences were normalized to input DNA, and then normalized to that at the non-enriched *ASI1* locus. Error bars indicate the SEM from at least three independent experiments. Statistical significance was analyzed by Student's t-test between WT and mutants. **p<0.01. *p<0.05. (H) Chromatin IP analysis of the occupancy of cohesion factor Eco1-13Myc at replication early origins ARS305 and ARS306 and at late origin ARS501 in α-F-arrested G1-phase WT and *bre1Δ* cells. IP signals at each ARS were normalized to input DNA, and then normalized to that at the non-enriched *ASI1* locus. Error bars represent the SEM from at least three independent experiments. *p<0.05, **p<0.01 by Student's t-test. (I) Western blotting analysis of whole-cell extract from log phase culture of WT and *bre1Δ* using anti-acetyl-Smc3 and Smc3-13Myc polyclonal antibody. Smc3-Ac and Smc3 levels were quantified by normalizing to Pgk1, the loading control. (J) Analysis of sister chromatid cohesion in WT, *lge1Δ*, bre1Δ, ctf4Δ, ctf18Δ, *ctf18Δ lge1Δ*, *ctf18Δ bre1Δ*, *ctf4Δ lge1Δ* and *ctf4Δ bre1Δ* cells after arrest in G2/M by Noc for 3 hr. The percentage of cells with two GFP signals is shown. At least 100 cells were scored. The results of three independent assays were averaged. The data for WT, *lge1Δ* and *bre1Δ* are the same as in *Figure 1F*, and are shown here for comparison with double mutants. The error bars correspond to the SEM from the mean value. **p<0.01 between WT and corresponding mutant by Student's t-test. NS stands for no significant difference among the indicated single and double mutants.

DOI: https://doi.org/10.7554/eLife.28231.006

The following figure supplements are available for figure 3:

**Figure supplement 1.** Bre1 is dispensable for cohesin loading.

DOI: https://doi.org/10.7554/eLife.28231.007

**Figure supplement 2.** Bre1 is important for recruiting cohesion factors to replication origins.

DOI: https://doi.org/10.7554/eLife.28231.008

single mutant of *ctf4Δ* and *ctf18Δ*, suggesting that *BRE1* and *LGE1* epistatically interact with *CTF4* and *CTF18*, and that *BRE1* and *LGE1* may function upstream of both *CTF4* and *CTF18* genetic pathways in cohesion, affecting their recruitment partially.

## Bre1 is required for the recruitment of replication factors that localize cohesion establishment factors

We hypothesized that Bre1 may signal to replication factors upstream of Ctf4 and Ctf18 to affect their localizations. Indeed, Bre1-mediated H2Bub1 is required for the association of some replisome proteins (Polα, Polε and RPA) (*Sun et al., 2016*) and replisome progression complex (RPC) components (Psf2 and Spt16) (*Gambus et al., 2006*) with early origins, and for stable replication fork progression (*Trujillo and Osley, 2012*). Psf2 is required for Ctf4 localization at origin of replications (*Gambus et al., 2006*), and in turn, Ctf4 is required for Polα localization (*Gambus et al., 2009*; *Zhu et al., 2007*). Thus, we attempted to search for replication factors more upstream than Psf2 that are regulated by Bre1. Yet, H2Bub1 is dispensable for the loading at origins of the most upstream origin recognition complex (ORC) component Orc2(*Trujillo and Osley, 2012*). Previous studies showed that both the loading of GINS, Polα and Ctf4 onto chromatin and CMG helicase activation depends on replication initiation and elongation factor Mcm10 (*Perez-Arnaiz et al., 2016*; *Quan et al., 2015*; *Ricke and Bielinsky, 2004*; *Zhu et al., 2007*). Therefore, we asked whether Bre1 affects the recruitment of Mcm10.

We verified that Bre1, like H2Bub1, is required for the association of Psf2 and Polα with chromatin and origins (*Trujillo and Osley, 2012*), and tested whether Mcm10 recruitment is also affected. Chromatin spreads showed that Psf2, Polα and Mcm10 associate with chromatin in G1, S and G2/M phases in WT cells (*Figure 4A–C* and *Figure 4—figure supplement 1D–F*), consistent with previous reports (*Falconi et al., 1993*; *Gambus et al., 2006*). However, the levels of Psf2, Polα and Mcm10 associated with chromatin in *bre1Δ* cells in G1 and S phases were significantly reduced, whereas those in G2/M phase was not affected. The reduced chromatin association of Psf2, Polα and Mcm10 in *bre1Δ* cells was not the consequence of reduced protein expression levels, as confirmed by western blotting analysis (*Figure 4—figure supplement 1A–C*). ChIP-qPCR further showed the occupancy of Psf2 at early replication origins in *bre1Δ* cells in S phase, the occupancy of Mcm10 at early origins in G1 and S phase, and the occupancy of Mcm10 at early origin-flanking regions in S phase were all significantly reduced, but that at a late origin their occupancy levels remained low but comparable to those in WT and *bre1Δ* (*Figure 4D–G*) (*Sekedat et al., 2010*). Interestingly, co-immunoprecipitation showed that Bre1 interacts weakly with Mcm10 (*Figure 4—figure supplement 1G*).

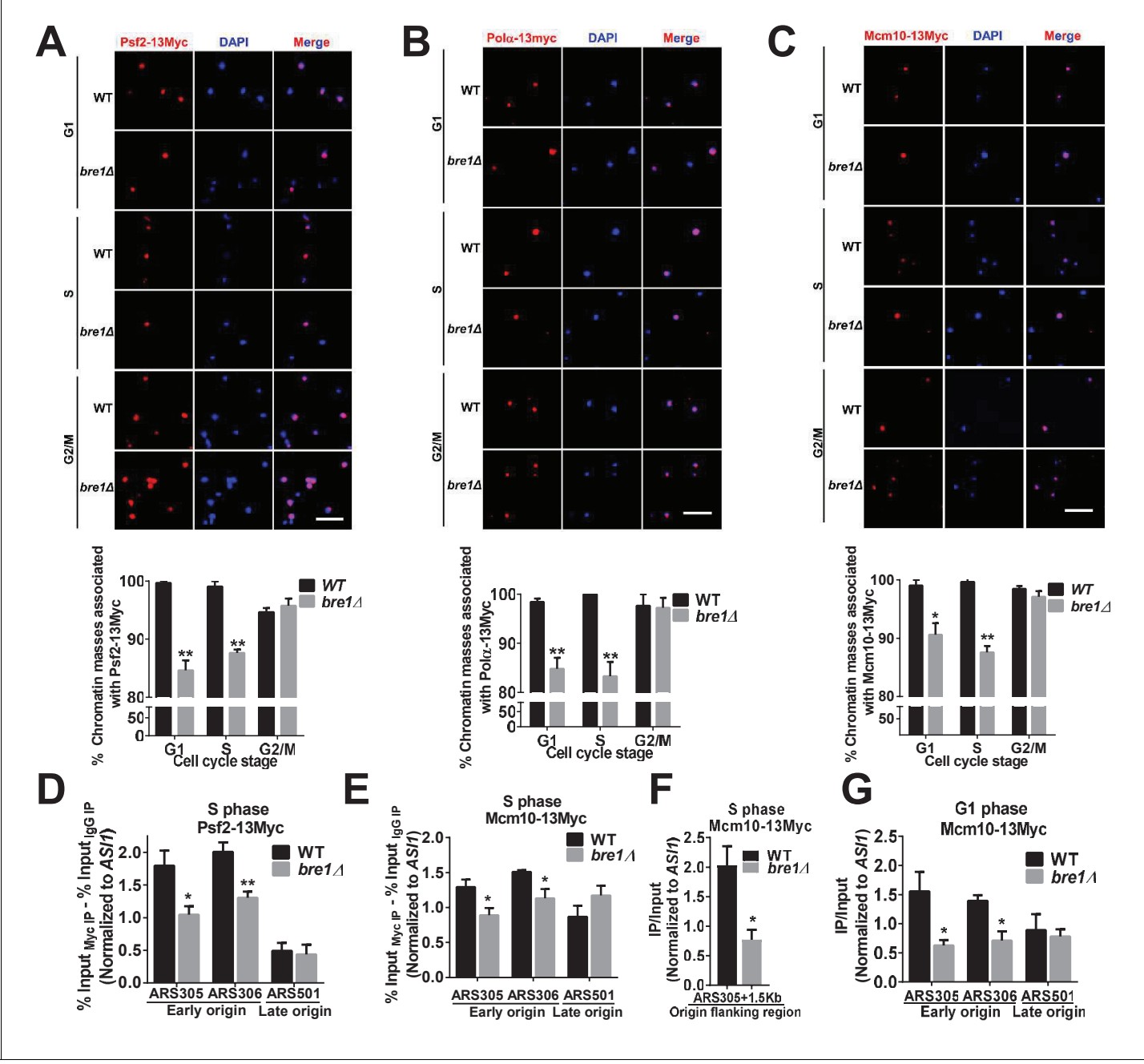

**Figure 4.** Bre1 recruits replication factors Psf2, Polα, and Mcm10 to chromatin and early replication origins. (A–C) Psf2-13Myc (A), Polα−13Myc (B) and Mcm10-13Myc (C) detected in chromatin spreads in wild-type (WT) and *bre1Δ* cells at indicated cell-cycle-arrested stages (G1, S or G2/M phase arrested with α-F, HU or Noc for 3 hr, respectively). The frequencies of chromatin-associated Psf2-13Myc, Polα−13Myc and Mcm10-13Myc were quantified. At least 100 chromatin masses were scored for each sample in each experiment. Error bars indicate SEM from at least three independent experiments. **p<0.01, *p<0.05 by Student's t-test. Scale bar: 1 μm. (D–F) Chromatin immunoprecipitation analysis of the occupancy of replication factors Psf2-13Myc (D) and Mcm10-13Myc (E and F) at replication early origins ARS305 and ARS306, late origin ARS501 (D and E) and origin flanking region ARS305 + 1.5 kb (F) in HU-arrested S-phase WT and *bre1Δ* cells. Immunopreciptiation (IP) signals at each ARS were normalized to input DNA, and then normalized to the signal at the non-enriched *ASI1* locus. Error bars represent the SEM from at least three independent experiments. **p<0.01, *p<0.05 by Student's t test. (G) Chromatin immunoprecipitation analysis of the occupancy of replication factor Mcm10-13Myc at replication early origins ARS305, ARS306 and late origin ARS501 in α-F-arrested G1 phase WT and *bre1Δ* cells. IP signals at each ARS were normalized to input DNA, and then normalized to the signal at the non-enriched *ASI1* locus. Error bars represent the SEM from at least three independent experiments. *p<0.05 by Student's t test.

DOI: https://doi.org/10.7554/eLife.28231.009

*Figure 4 continued on next page*

*Figure 4 continued*

The following figure supplement is available for figure 4:

**Figure supplement 1.** Bre1 does not affect the expression of replication factors Psf2, Polα and Mcm10.
DOI: https://doi.org/10.7554/eLife.28231.010

However, Bre1 is dispensable for Mcm10 diubiquitination (Mcm10[Ub]$_2$) (*Figure 4—figure supplement 1H*). Collectively, we demonstrated that Bre1 plays a role in regulating the localization of an upstream replication factor, Mcm10, which is important for CMG activation, replication initiation at origins, replication fork progression, and recruitment of replication-coupled cohesion establishment factors.

## Bre1's catalytic RING domain and target H2Bub1 are required for cohesion establishment by recruiting cohesion establishment factor Ctf4 and replication factor Mcm10

E3 ubiquitin ligase Bre1 is known to function with E2 ubiquitin conjugating enzyme Rad6 to mono-biquitinate histone H2B at K123 (H2Bub1) through the conserved RING domain of Bre1 (*Hwang et al., 2003*; *Robzyk et al., 2000*). We tested whether Rad6 plays a role in cohesion as Bre1 does, and whether Bre1 functions in cohesion through its ubiquitin ligase activity on its target H2Bub1. We examined *rad6Δ*, and made use of the Bre1 RING domain truncation mutant *bre1-RINGΔ*, which lacks E3 activity and cannot ubiquitinate H2B as shown in a prior study (*Hwang et al., 2003*) (*Figure 5A*). In addition, we constructed a *H2B* mutant that cannot be ubiquitinated. To do this we used K123R point mutation in one of the *H2B* genes, *HTB1*, and deletion of another *H2B* gene *HTB2* (*htb1K123R htb2Δ*) as described in prior work (*Robzyk et al., 2000*). We then performed the cohesion assay in G2/M-arrested cells by adding nocodazole for 3 hr. The cohesion defect in *rad6Δ* (19.6% of cells), *bre1-RINGΔ* (18.0%) and *htb1-K123R htb2Δ* (17.9%) occured at frequencies comparable to that in *bre1Δ* in nocodazole (18.8%) (*Figure 5B*) and that in *bre1Δ* in 60–75 min post G1 arrest and release (20.2%, *Figure 1D*). On the other hand, *htb2Δ* showed a less frequent but significant cohesion defect (10.0% of cells), and Flag-*HTB1* (7.3%) showed low levels of cohesion defect comparable to those of WT controls (5.9%) (*Figure 5B*). These results suggest that Rad6 and Bre1-catalyzed H2Bub1 accounts for Bre1's function in sister chromatid cohesion. By contrast, deletion of the other known substrate of Bre1, *swd2Δ* (5.6%) (*Vitaliano-Prunier et al., 2008*), showed only a WT level of cohesion defect. The cohesion defect in the *bre1Δ lge1Δ* double mutant in nocodazole-arrested G2/M cells (~17.4% of cells) was similar to that in single *bre1Δ* or *lge1Δ* (18.8% or 18.3% of cells, respectively) (*Figure 5B*), consistent with Bre1 and Lge1 functioning together in a complex.

To distinguish the role of Rad6, the Bre1 RING domain and H2Bub1 in sister chromatid cohesion establishment in S phase versus maintenance in G2/M phase, we repeated the G1 arrest and release cohesion assay over a time course (*Figure 5C*, as in *Figure 1B and C*), and confirmed that the WT has a low percentage of premature sister chromatid separation up to 90 min after release from G1. On the other hand, *rad6Δ, bre1Δ, bre1-RINGΔ,* and *htb1-K123R htb2Δ* have progressively elevated frequencies of premature sister chromatid separation from 30 to 90 min after release from G1, shortly after DNA replication begins (*Figure 5—figure supplement 1A*), which are comparable to frequencies seen in cohesion establishment factor mutants such as *ctf18Δ* (*Hanna et al., 2001*).

Consistent with the cohesion defect shown by *bre1-RINGΔ* and *htb1K123R htb2Δ* (*Figure 5B and C*), we found that these two mutants also reduce the association of cohesion establishment factor Ctf4 and replication factor Mcm10 at early origins in S phase (*Figure 5D–G*). These results further suggest that the role of Bre1 in cohesion is through its catalytic RING domain and H2B monoubiquitination. As Bre1 stability has been shown to be affected by its RING domain's catalytic activity, and by its ubiquitination level at H2BK123 (*Wozniak and Strahl, 2014*), we checked the endogenously tagged Bre1 protein level in WT, *bre1-RINGΔ* and *htb1-K123R htb2Δ* cells. Consistent with previous findings, Bre1 protein level was reduced in the *bre1-RINGΔ* mutant. Thus, the effect of a *bre1-RINGΔ* mutation on cohesion could be due to either the loss of Bre1's E3 catalytic activity or reduced Bre1 level. However, Bre1 protein level was comparable in *htb1-K123R htb2Δ* and WT (*Figure 5—figure*

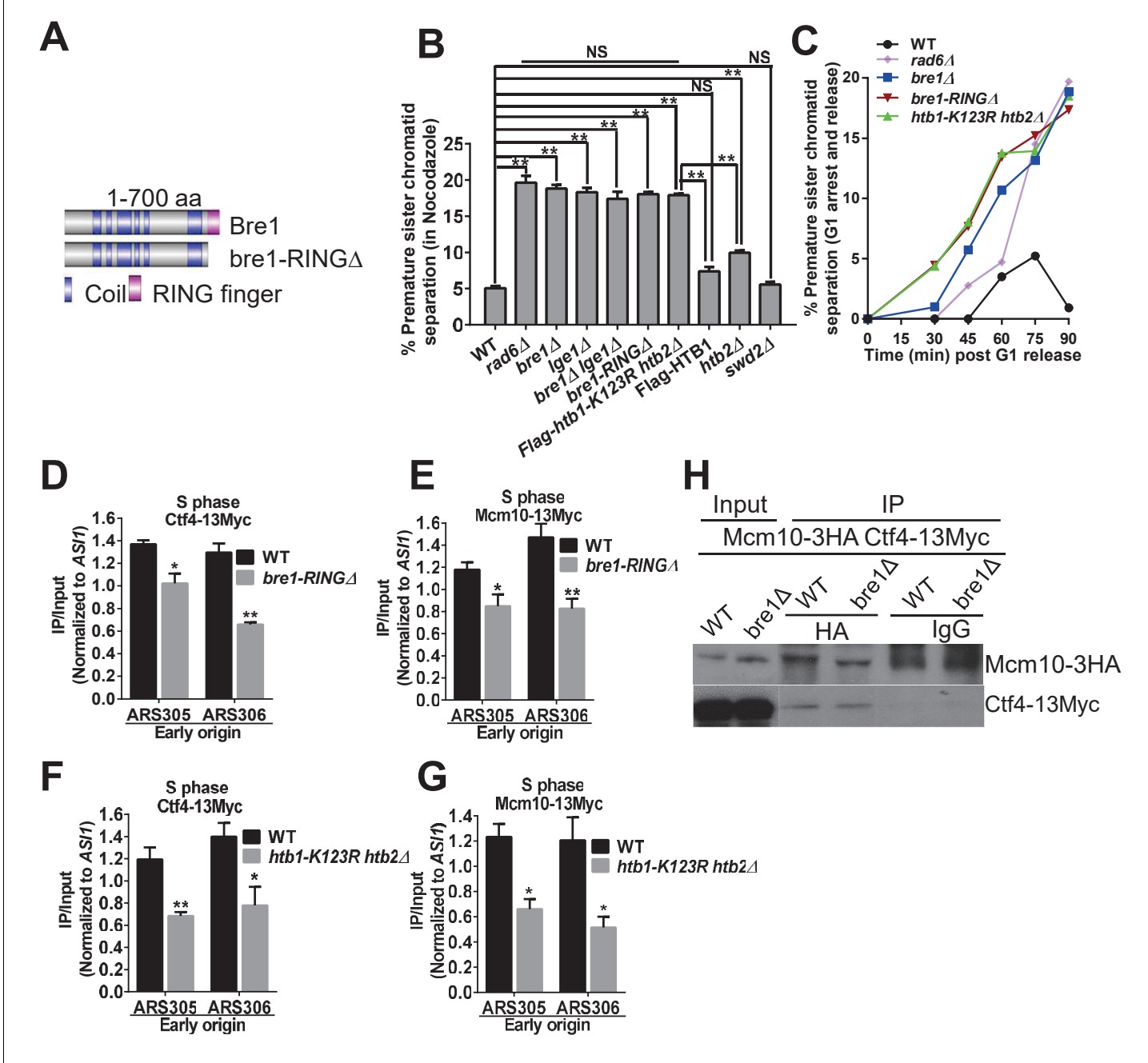

**Figure 5.** Bre1 RING domain and H2B monoubiquitination are important for cohesion establishment and the recruitment of replication and cohesion factors to early origins. (**A**) Schematic diagram of full-length Bre1 protein (700 amino acids) and the Bre1 mutant protein without the C-terminal RING domain (54 amino acids), bre1-RINGΔ, which is defective in H2B ubiquitination. (**B**) Analysis of sister chromatid cohesion in wild-type (WT) and mutant strains in nocodazole-arrested cells. WT, *rad6Δ, bre1Δ, lge1Δ, bre1Δ lge1Δ, bre1-RINGΔ, htb1-K123R htb2Δ*, Flag-*HTB1, htb2Δ*, and *swd2Δ* mutants were arrested in G2/M phase by nocodazole for 3 hr before being assessed by cohesion assay. At least 100 were scored for each strain in three independent experiments. Error bars represent SEM. Statistical significance was analyzed by Student's t-test between WT and mutants (\*\*p<0.01). NS represents no significant difference between WT and *swd2Δ* or Flag-tagged *HTB1* control, or among *rad6Δ, bre1Δ, lge1Δ, bre1Δ lge1Δ, bre1-RINGΔ*, and *htb1-K123R htb2Δ* mutants. (**C**) Analysis of sister chromatid cohesion in WT and *rad6Δ, bre1Δ, bre1-RINGΔ*, and *htb1-K123R htb2Δ* mutant strains by G1 arrest and release time-course assay. The WT strain has a low percentage of cells with premature sister chromatid separation from 0 to 90 min after release from G1, whereas *rad6Δ, bre1Δ, bre1-RINGΔ*, and *htb1-K123R htb2Δ* have progressively elevated occurences of premature sister chromatid separation from 30 min after release from G1. At least 100 cells were scored for each sample. (**D** and **E**) Chromatin immunoprecipitation analysis of the occupancy of replication factors Ctf4-13Myc (**D**) and Mcm10-13Myc (**E**) in WT and the *bre1-RINGΔ* mutant at replication early origins ARS305 and ARS306 in HU-arrested S phase. Immunoprecipitation (IP) signals at each ARS were normalized to input DNA, and then normalized to signals at the

*Figure 5 continued on next page*

*Figure 5 continued*

non-enriched *ASI1* locus. Error bars represent the SEM from at least three independent experiments. \*\*p<0.01, \*p<0.05 by Student's t-test. (F and G) Chromatin IP analysis of the occupancy of replication factors Ctf4-13Myc (F) and Mcm10-13Myc (G) in WT and the *htb1-K123R htb2Δ* mutant at replication early origins ARS305 and ARS306 in HU-arrested S phase. IP signals at each ARS were normalized to input DNA, and then normalized to that at the non-enriched *ASI1* locus. Error bars represent the SEM from at least three independent experiments. \*\*p<0.01, \*p<0.05 by Student's t-test. (H) Co-immunoprecipitation analysis of the interaction between Ctf4 and Mcm10 in wild-type and *bre1Δ* cells. Whole cell extracts prepared from Mcm10-3HA Ctf4-13Myc WT and *bre1Δ* cells were precipitated with anti-HA and anti-IgG antibodies. Immunoprecipitates were probed with anti-HA or anti-Myc antibody, respectively.

DOI: https://doi.org/10.7554/eLife.28231.011

The following figure supplements are available for figure 5:

**Figure supplement 1.** Cell cycle progression in wild-type, *rad6Δ*, *bre1Δ*, *bre1-RINGΔ* and *htb1-K123R htb2Δ*.

DOI: https://doi.org/10.7554/eLife.28231.012

**Figure supplement 2.** Bre1 protein levels in *bre1-RINGΔ* and *htb1-K123R htb2Δ* mutants.

DOI: https://doi.org/10.7554/eLife.28231.013

*supplement 2A and B*), suggesting that Bre1-mediated H2B monoubiquitination, but not Bre1 protein level, accounts for the cohesion defect in *htb1-K123R htb2Δ*.

As Bre1 and H2Bub1 are required for the recruitment of Ctf4 and Mcm10, and as Ctf4 interacts with Mcm10 (*Wang et al., 2010*), we checked whether *bre1Δ* disrupts this interaction. By performing co-immunoprecipitation between Ctf4 and Mcm10, we found that *bre1Δ* mutation does not affect the interaction of Ctf4 with Mcm10 (*Figure 5H*).

## Discussion

### Bre1-mediated H2Bub1 marks replication origins and recruits DNA replication factors in alpha-factor and HU arrest, facilitating the association of cohesion establishment factors and cohesion establishment in S phase

The conserved E3 ubiquitin ligase Bre1, responsible for H2B monoubiquitination, contributes to structural chromosome integrity, which is well evidenced by its characterized roles in DNA replication, transcription, DNA damage response and repair processes through modulating nucleosome dynamics and histone crosstalk signaling. However, the underlying cause of whole chromosome instability (CIN) in *BRE1*-deletion mutants is not fully understood. In this study, we have identified a novel role for Bre1, its interacting partner Lge1and H2Bub1, catalyzed by the RING-finger domain of Bre1, in precise chromosome segregation and sister chromatid cohesion. Whereas Bre1 is non-essential, and so the deletion mutant is viable, our degron mutant together with assays of cohesion establishment and replication factors recruitment help to pinpoint the timing of Bre1's function in cohesion to G1 and S phases. Although Bre1 is dispensable for the loading of cohesin subunits Scc1 and Smc3 onto chromatin, it facilitates the recruitment of cohesion establishment factors Ctf4, Ctf18 and Eco1 to chromatin and to early origins in S phase to promote Smc3 acetylation. It is known that H2Bub1 is required to regulate the occupancy of active Mcm4, Cdc45, Psf2 and Polα at the early origins in S phase, but this protein is not required for the localization of ORC, inactive Mcm4 and Cdc45 in G1 phase, as shown in a prior study (*Trujillo and Osley, 2012*). Here we identified a further upstream, essential replication initiation and elongation factor, Mcm10, which is important for CMG (Cdc45-Mcm2-7-GINS) helicase assembly and activation (*Perez-Arnaiz et al., 2016*) and whose recruitment to chromatin and early origins is at least partially affected by non-essential Bre1.

These findings are compatible with a model in which Bre1 localizes to origins and monoubiquitinates H2B (*Trujillo and Osley, 2012*), which acts as an upstream epigenetic mark to signal the recruitment of both the replication factors (Mcm10, Psf2, and Polα) and the cohesion establishment factors (Ctf4, Ctf18 and Eco1) to origins (*Figure 6*). Mcm10, Psf2 and Polα each interacts with Ctf4, an RPC component that is also required for sister chromatid cohesion (*Gambus et al., 2006*, *2009*; *Simon et al., 2014*; *Tanaka et al., 2009*; *Wang et al., 2010*; *Wittmeyer and Formosa, 1997*; *Zhu et al., 2007*). The partially reduced level of chromatin-associated replication factors (Psf2 and Mcm10) in *BRE1* null mutant affects the localization of Ctf4, which in turn affects the localization of

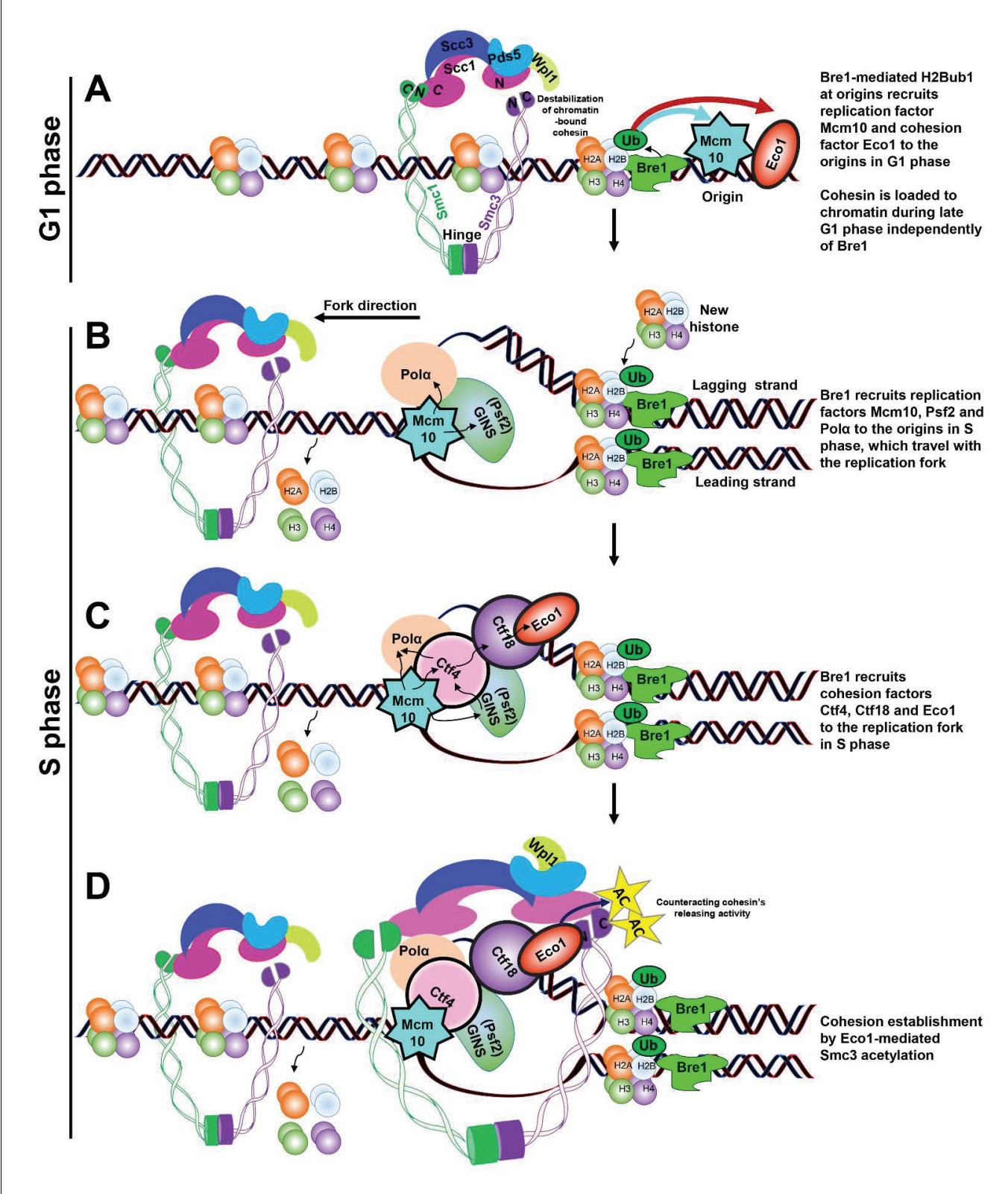

**Figure 6.** Schematic model of Bre1's role in replication-coupled cohesion establishment. (**A**) A Bre1- and H2Bub1-marked origin (***Trujillo and Osley, 2012***) partially facilitates the occupancy of replication factor Mcm10 and cohesion factor Eco1 at the origins in G1 phase (as indicated by the blue and red arrows). Cohesin complexes are loaded onto chromatin before the onset of DNA replication (***Ciosk et al., 2000***) independently of Bre1, but the cohesin association on chromatin is destabilized by Wpl1. (**B**) In S phase, the Bre1- and H2Bub1-marked origin partially facilitates the recruitment of

*Figure 6 continued on next page*

*Figure 6 continued*

replication factors Mcm10, Psf2 and Polα to the origins, which travel with the replication fork (*Trujillo and Osley, 2012*). (C) Through these replication factors, the Bre1- and H2Bub1-marked origin partially facilitates the recruitment of cohesion factors Ctf4, Ctf18 and Eco1 to the origins and replication forks. (D) Bre1 promotes Eco1-mediated Smc3 acetylation (as indicated by the yellow stars and black arrows) at the replication fork to facilitate and stabilize cohesion establishment by counteracting Wapl1's activity in releasing cohesin.

DOI: https://doi.org/10.7554/eLife.28231.014

Ctf18, and thus Eco1, leading to reduced Smc3 acetylation and resulting in defective cohesion establishment. Surprisingly, PCNA's association at early origins was unaffected in the absence of H2Bub1 (*Trujillo and Osley, 2012*). This could be because the partial reduction of Ctf18's chromatin association is not severe enough, or because PCNA can associate with origins through a Ctf18-independent pathway. Bre1 and H2Bub1 play a role in the progression of replication forks, in which these replication and cohesion establishment proteins travel together to allow cohesion establishment on replicated sister chromatids. Our findings provide new supporting evidence for the proposed model in which the establishment of cohesion is coupled with DNA replication (*Lengronne et al., 2006*; *Terret et al., 2009*). Whether the replication fork could slide through the existing cohesin ring, and the exact orientation of cohesin rings, is still unclear. However, as Bre1 is non-essential and as in its absencesome level of replication factors and cohesion factors still localize to chromatin and replication origins, additional factors in independent pathways possibly also contribute to the recruitment of these factors to origins.

## Potential role of H2Bub1 with regard to replicated DNA and DSBs in cohesion establishment

Our data suggest that Bre1's function in recruiting replication factors to origins is important for its function in cohesion establishment. However, H2Bub1 has also been shown to function in nucleosome reassembly and is retained in newly replicated DNA (*Trujillo and Osley, 2012*). Whether H2Bub1 level, which doubled in replicated DNA (*Trujillo and Osley, 2012*), and H3K56Ac, an epigenetic mark for newly synthesized H3 which assembles upon DNA replication (*Kaplan et al., 2008*), could play a direct role in signaling successful replication to the cohesion establishment pathway is unexplored. In addition, how Bre1 is temporally and spatially regulated, for example during its recruitment to origins in G1 phase and its maintenance in sisters in S phase, needs to be addressed in the future. As does the question of whether the deubiquitinisation of H2bub1 at a later stage in the cell cycle is relevant to cohesion function.

As H2Bub1 is important for DSB repair (*Moyal et al., 2011*; *Nakamura et al., 2011*; *Yamashita et al., 2004*), it will be interesting to investigate whether it is important for DSB-induced cohesion establishment. The human homologs of yeast *BRE1*, *RNF20* and *RNF40* are mutated and misregulated in different types of cancers. Whether defective Rnf20 or Rnf40 leads to defective cohesion and CIN in human cells, and whether this contributes to tumorigenesis initiation and progression, is worth pursing in order to reveal the genetic basis of CIN in cancers.

## Materials and methods

### Yeast strains, plasmids and media

The gene deletion strains and strains that expressed 3HA- or 13Myc-tagged proteins were generated by the PCR-based gene deletion and modification methods as described before (*Longtine et al., 1998*). The yeast strains and plasmids used in this study are listed in *Supplementary files 1* and *2*, respectively. To generate *htb-K123R* mutant strains, a fragment consisting of Flag-*HTB1*-K123R with the *URA3* marker was amplified from WYYp30 and integrated at the endogenous *HTB1* locus of YPH1343 by homologous recombination, in addition to deleting *HTB2*. Flag tagged HTB1 from WYYp19 was integrated at the endogenous *HTB1* locus of YPH1343 by homologous recombination as a WT control. To make the *BRE1*-degron strain, plasmid WYYp74 was used to amplify the fragment AID*−9Myc (AID*: minimum functional size region [71–114 amino acids] of full-length AID [229 amino acids]) with *KanMX6* marker at the 3′ end, which was

transformed into YPH1343, generating WYYY250. In addition, the plasmid containing OsTIR1 (pNHK53) was linearized with StuI and integrated into WYYY250, creating WYYY326.

Yeast cells were routinely grown in YPD (1% yeast extract, 2% peptone, and 2% dextrose)-rich media at 30°C. Synthetic complete (SC) medium lacking a specific amino acid was used for selection. An SC with limiting adenine plate was prepared as described previously (*Hieter et al., 1985*). A final concentration of 400 µg/ml of G418 (Cat#: G4185, Formedium, England) antibiotic was used for the selection of gene deletions and epitope tagging with *KanMX6* marker.

## Chromosome transmission fidelity (CTF) assay

The CTF assay was performed as described previously (*Spencer et al., 1990*; *Yuen et al., 2007*). Briefly, WT and gene-deletion cells containing an *ade2-101* (*ochre*) mutation and a *SUP11*-marked chromosome III fragment (CFIII) were picked from plates selecting for the CFIII (SC-URA), and then plated onto minimal (SD) medium non-selective for the CFIII (SC with 20% limiting adenine [10 µg/ ml]) at a density of ~200 colonies per plate. The plates were incubated at 30°C for 2–3 days and then placed at 4°C to facilitate red pigment development. Cells containing the CFIII were white, whereas those that had lost the CFIII were red. Therefore, a white-and-red sectored colony was observed if the CFIII is lost in some mitoses during the formation of the colonies. Colonies that were at least half red were considered as having a chromosome loss event during the first division. The loss frequency of the CFIII was calculated as the ratio of the number of over half-red colonies to the total number of colonies. At least 2000 cells were scored in each experiment, and three independent experiments were performed.

## G1 arrest and release cohesion assay and chromosome segregation assay

G1 arrest and release cohesion assays and chromosome segregation assays were carried out as previously reported (*Straight et al., 1996*) with minor modifications. Basically, early log phase cultures with optical density (OD) at 600 nm around 0.2 to 0.4 were collected, washed with water and arrested in the G1 phase with 5 µg/ml alpha-factor (α-F) for 3 hr. Cells were washed with water and released into YPD medium. α-F was added back to the culture at 60 min post G1 release to restrict cells in the next cell cycle at G1. Samples were collected every 15 min for fluorescence-activated cell sorting (FACS) analysis. Cohesion assays were performed on large-budded cells at 60–75 min after release from G1 arrest, at which time the majority of cells reached G2/M phase by FACS and budding index, or at 15 min-intervals between 30–90 min after release from G1 arrest. Chromosome segregation assays were carried out on unbudded cells at 120–150 min after release from G1 arrest, at which time most cells had completed cytokinesis and had no bud, and the FACS profiles showed that the majority of cells were in G1 phase. Cells at G2/M phase and G1 phase were fixed with freshly prepared 4% paraformaldehyde at room temperature for 15 min, followed by a wash with SK buffer (a 1% potassium acetate [Kac]−1M sorbitol solution) and centrifugation at 2000 rpm for 2 min. Pellets were resuspended in SK buffer for cohesion assessment.

## Nocodazole-arrested cohesion assay

The nocodazole (Noc)-arrested cohesion assay was performed as reported previously with slight modification (*Hanna et al., 2001*). Early log phase cells were harvested and washed with water and released into YPD medium containing 15 µg/ml Noc for 3 hr. Samples were collected, fixed and resuspended in SK buffer as mentioned above for the G1 arrest and release cohesion assay.

## Visualization of Lac operator staining in yeast

Cells were imaged on a Carl Zeiss LSM 710 NLO confocal laser scanning microscope using an EC Plan-Neofluar 40x/1.30 Oil Ph3 M27 oil objective and a conventional FITC excitation filter. Z-stacked images were acquired (six z-sections were acquired at 1 µm intervals). In cohesion experiments, at least 100 large budded cells were scored as containing one or two GFP foci. In the chromosome segregation assays, at least 100 G1 cells were scored, and data were averaged from at least three independent experiments.

## Bre1-AID*−9Myc time-course degradation assay

The Bre1-AID*−9Myc cells were arrested in G1 with 5 µg/ml alpha factor (α-F) for 3 hr and washed with water before splitting into three cultures. The first and second split cultures were released into YPD containing 5 µg/ml α-F to maintain G1 phase or 0.2 M hydroxyurea (HU) to arrest cells in S phase, in the presence of 1 mM auxin for 2 hr. The third G1 culture was released into YPD consisting of 0.2 M HU to arrest cells in S phase for 2 hr, and subsequently released into 15 µg/ml nocodazole (Noc)-containing YPD to arrest cells into G2/M phase with 1 mM auxin for 2 hr. Upon auxin addition, samples were collected every 15 min for FACS analysis as described in a previous study (*Hanna et al., 2001*) and for western blotting analysis of Bre1 protein level using anti-Myc antibody and Pgk1 as the loading control. Bre1-AID*−9Myc protein levels were quantified by Image J software. The normalized signal density value for each sample band at an indicated time point was calculated as the ratio of the relative density of each sample lane (after subtracting background) over the relative density of the Pgk1 loading control for the same lane (*Miller, 2010*). The normalized protein amounts relative to that before auxin addition were plotted in graphs.

## Auxin-induced Bre1-AID*−9Myc degradation and cohesion assays

Auxin was added to the cell cultures arrested at the indicated cell-cycle stages to induce the degradation of Bre1 as described below. (1) For the no auxin-induced degradation control, early log-phase cells containing Bre1-AID*−9Myc were arrested in G1 with 5 µg/ml alpha-factor (α-F) for 3 hr before releasing into YPD medium containing 5 µg/ml α-F for another 2 hr to maintain G1 arrest. Then, G1 phase cells were released into hydroxyurea (HU)-containing media to arrest cells in S phase for 2 hr. Finally, S phase cells were released into Noc-containing media to arrest cells in G2/M phase for 3 hr. (2) For Bre1 degradation in all the stages, cell-cycle arrest procedures were the same as described in (1), except that cells in HU-containing medium were arrested for 4 hr instead of 2 hr due to a delayed G1-S transition and progression in S phase in the presence of auxin in G1. Bre1 degradation at each stage was induced by the addition of 1 mM auxin to the same medium. (3) For Bre1 degradation in G1 and S, procedures were the same as in (2) except in the last step, when S phase cells were released into Noc-containing medium without auxin. (4) For Bre1 degradation in G1 phase, procedures were the same as in (3) except that G1-arrested cells were released into HU-containing medium without auxin. (5) For Bre1 degradation in S phase, procedures were the same as in (1) except that G1-arrested cells were released into HU-containing medium with auxin. (6) For Bre1 degradation in G2/M phase, procedures were the same as in (1) except that S-phase-arrested cells were released into Noc-containing medium with auxin. (7) For Bre1 degradation in S and G2/M phases, procedures were the same as in (1) except that auxin was added into both HU- and Noc-arrested cells. Samples were collected every 30 min for flow cytometry analysis of DNA content and western blotting analysis for quantification of Bre1 protein levels as described above. In all the cases, G2/M-phase cells were collected after Noc arrest for 4 hr for cohesion assessment.

## Yeast cell lysate preparation and western blotting

Yeast whole cell extracts were prepared using the trichloroacetic acid (TCA) precipitation method as described by the Dohlman lab (http://www.med.unc.edu/~dohlmahg//TCA.html). The protein concentration was determined with the Bio-Rad DC protein assay kit. Equal amounts of protein samples were boiled in 4xSDS-PAGE sample buffer and subjected to SDS-PAGE gel, before being transferred to Immobilon PVDF membrane (CAT#: 1620177, Millipore, Ireland). The membrane was blocked in TBST (20 mM Tris-HCl, 125 mM NaCl, 0.1% Tween-20) containing 5% non-fat dry milk for 1 hr at room temperature before probing using antibodies against HA (12CA5, 1:1000, Cat#: 11583816001, Roche, Germany), Myc (9E10, 1:2000, Cat#: 05–419, Millipore, Billerica MA, USA), Rad53p (1:2000, Cat#: ab104232, Abcam, San Francisco, CA, USA), Pgk1 (1:6000, Cat#: ab113687, Abcam, Frederic, MD, USA), acetyl-Smc3 (1: 500, a gift from the Dmitry Ivanov Lab), Flag (M2, 1:2000, Cat#: F1804, Sigma-Aldrich, St. Louis, MO, USA) or ubiquitin (P4G7-H11, 1:1000, Cat#: ab90376, Abcam, San Francisco, CA, USA) and incubated overnight at 4°C. The membrane was washed three times with TBST for 10 min each time, and subsequently incubated with the secondary antibody (Goat polyclonal Secondary Antibody to Rabbit IgG H and L [HRP, 1:100000, Cat#: ab97051, Abcam, San Francisco, CA, USA or Goat polyclonal Secondary Antibody to Mouse IgG H and L [HRP, 1:100000, Cat#: ab97023, Abcam, San Francisco, CA, USA]) for 30 min at room

temperature. After washing, the blots were detected using the Amersham ECL Select western blotting detection reagent (GE Healthcare Life Sciences, UK) and developed using X-ray film.

## Chromatin spread

Early log-phase cells were arrested in G1, S or G2/M phase with 5 µg/ml α-F, 0.2 M HU or 15 µg/ml Noc, respectively, for 3 hr. Chromatin spread was performed on slides as described previously (*Grubb et al., 2015*; *Rockmill et al., 2009*). The slides were blocked with 300 µl 1% bovineserum albumin (BSA)/PBS in a moist chamber for 15 min at room temperature and subsequently incubated with 100 µl of primary antibody against HA (1:200, Cat#: sc-57592, Santa Cruz Biotechnology, Dallas, TX, USA) or Myc (1:200, Cat#: sc-40, Santa Cruz Biotechnology, Dallas, TX, USA) in 1% BSA/PBS at room temperature for 2 hr. Samples were then washed three times with PBS, and incubated with CY3-conjugated goat anti-mouse secondary antibody (1:500, Cat#: 115-166-062, Jackson Immunoresearch Labs, West Grove, PA, USA) in 1% BSA/PBS at room temperature for 2 hr. Finally, the slides were stained with 4', 6-diamidino-2-phenylindole (DAPI) (1 µg/ml) for 5 min, washed with PBS, and mounted with mounting media. The slides were then processed for immunostaining. Indirect immunofluorescence was observed using a Carl Zeiss LSM 710 NLO confocal laser scanning microscope with a 40x/1.4 NA oil objective and a conventional FITC excitation filter. The percentage of chromatin masses associated with epitope-tagged protein was calculated as the ratio of the number of chromatin masses with epitope tag signals over the number of chromatin masses with DAPI signals. At least 100 chromatin masses were scored, and data were averaged from at least three independent experiments.

## Chromatin immunoprecipitation (ChIP) and quantitative PCR

ChIP was carried out according to the methods used in previous studies with slight modifications (*Wahba et al., 2013*; *Zakari et al., 2015*). In brief, 100 ml early-log-phase cells were arrested in S phase with 0.2 M HU for 3 hr at 30°C (*Ricke and Bielinsky, 2004*). 1% formaldehyde was used for crosslinking for 20 min at room temperature. Spheroplasts were prepared as described for the chromatin spread assay. Spheroplasts were resuspended in SDS lysis buffer (1% SDS, 10 mM EDTA, 50 mM TRIS, pH 8.1) with proteinase K inhibitor and sonicated by an AFA focused-ultrasonicator (Covaris): 10% duty cycle, 75 watts intensity of peak incident power, 200 cycles per burst, 4 min to obtain sheared DNA fragments of 250–1000 bp in length (average 500 bp). Crosslinked proteins were immunoprecipitated with monoclonal anti-HA antibodies (12CA5) or anti-Myc antibodies (9E10), as well as Mouse IgG (Cat: 12–371, Millipore, Temecula, CA, USA) as a control for specificity, overnight at 4°C. The immune complexes were harvested by the addition of 50 µl of protein A dynabeads (Cat: 10001D, Thermo Fisher Scientific Inc, Norway). Formaldehyde crosslinks were reversed by incubation at 65°C for 5 hr, followed by protease K treatment at 42°C for 1–2 hr, and purification of recovered DNA was achieved using the ChIP DNA Clean and Concentrator kit (Zymo Research Corporation, Cat#: D5205, Irvine, USA). The purified DNA was subjected to quantitative real-time PCR using the StepOnePlus Real-Time PCR System (ABI). Primers for origins used were the same as those used previously (*Trujillo and Osley, 2012*). The % Input for each IP ($2^{-\Delta Ct}$) at origins was calculated, and the specific enrichment (% Input Ab IP – % Input IgG IP, according to SABiosciences ChIP-qPCR Array data analysis method, http://www.sabiosciences.com/chippcrarray_data_analysis.php) was further normalized to the specific enrichment at the *ASI1* locus (40 kb from the nearest origin). The normalized ratios of three independent IP experiments, each with duplicates or triplicate qPCR reactions for each primer set, were averaged and plotted on each graph as the relative enrichment of proteins.

## Co-immunoprecipitation assay

Immunoprecipitation was performed as described previously with modifications (*Gerace and Moazed, 2014*): yeast whole-cell extracts at early log phase were prepared by bead-beating in lysis/IP buffer (50 mM Tris-HCl [pH 7.5], 100 mM NaCl, 5 mM EDTA [pH 8.0], 0.1% NP40, 1 mM DTT and protease inhibitors) from 100 mL of log-phase culture. Extracts were precipitated with anti-HA (12CA5) or anti-Myc (9E10) antibody at 4°C overnight, followed by incubation with protein A dynabeads for 2 hr at room temperature. Dynabeads were then washed three times with lysis buffer. Associated proteins were eluted by incubating beads with SDS sample buffer for western blotting.

## Reverse transcriptase-quantitative PCR (RT-qPCR)

10 mL log-phase cells (OD600 ~0.5) were used for RNA isolation as described previously (*Ares, 2012*). A total of 1 µg RNA was used for cDNA synthesis using the ThermoScript RT-PCR System. cDNA was analyzed by RT-qPCR using primers specific for the cell cyclin genes *CLN2* and *CLB5*, cohesion genes *SCC1*, *SMC3*, *CTF4*, *CTF18* and *ECO1*, and actin gene *ACT1* (primer sequences are shown in *Supplementary file 3*). PCR was carried out using Applied Biosystems SYBR Green PCR Master Mix. For each gene, relative expression levels were calculated by the comparative CT method (StepOne software v2.3, from Applied Biosystems) obtained by qPCR assays of cDNA samples. Finally, *CLN2*, *CLB5*, *SCC1*, *SMC3*, *CTF4*, *CTF18* and *ECO1* expression levels were normalized to that of *ACT1*.

## Statistical analysis

Data are expressed as the mean ± standard error of the mean (SEM) from the number of independent experiments indicated in the figure legends. Student's t-test was used to analyze statistical significance.

## Acknowledgements

We thank Mary Ann Osley, Helle D Ulrich, Masato Kanemaki and Brian D Strahl for providing plasmids, and Dmitri Ivanov for providing antibody. We thank Abby Mak and Albert Au for technical assistance. We are grateful to the University of Hong Kong School of Biological Sciences Central Facilities for the use of their microscopy facility.

## Additional information

### Funding

| Funder | Grant reference number | Author |
|---|---|---|
| Hong Kong Research Grant Council | General Research Fund 17126714 | Karen Wing Yee Yuen |

The funders had no role in study design, data collection and interpretation, or the decision to submit the work for publication.

### Author contributions

Wei Zhang, Conceptualization, Data curation, Formal analysis, Validation, Investigation, Visualization, Methodology, Writing—original draft, Writing—review and editing; Clarence Hue Lok Yeung, Data curation, Investigation, Acquisition and analysis of data for Bre1-AID-degron assay; Liwen Wu, Data curation, Investigation, Acquisition and analysis of data for CTF assay; Karen Wing Yee Yuen, Conceptualization, Resources, Formal analysis, Supervision, Funding acquisition, Validation, Investigation, Visualization, Methodology, Writing—original draft, Project administration, Writing—review and editing

### Author ORCIDs

Wei Zhang http://orcid.org/0000-0001-8944-684X
Karen Wing Yee Yuen http://orcid.org/0000-0002-2139-5465

### Decision letter and Author response

Decision letter https://doi.org/10.7554/eLife.28231.019
Author response https://doi.org/10.7554/eLife.28231.020

## Additional files

### Supplementary files

• Supplementary file 1. Yeast strains used in this study.

DOI: https://doi.org/10.7554/eLife.28231.015

• Supplementary file 2. Yeast plasmids used in this study.
DOI: https://doi.org/10.7554/eLife.28231.016

• Supplementary file 3. Sequences of primers used in RT-qPCR experiments in this study.
DOI: https://doi.org/10.7554/eLife.28231.017

• Transparent reporting form
DOI: https://doi.org/10.7554/eLife.28231.018

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
