## [Decision Letter]

[Editors’ note: a previous version of this study was rejected after peer review, but the authors submitted for reconsideration. The first decision letter after peer review is shown below.]

Thank you for choosing to send your work, "E3 ubiquitin ligase Bre1 couples sister chromatid cohesion establishment to DNA replication in *Saccharomyces cerevisiae*", for consideration at *eLife*. Your initial submission has been assessed by a Senior Editor in consultation with a member of the Board of Reviewing Editors and two experts in the area. Although the work is of interest, we regret to inform you that the findings at this stage are too preliminary for further consideration at *eLife*.

As you will see from the reviewers' comments, both the BRE and the reviewers felt that the data presented in your study is open to a number of different interpretations and lacks many important controls. It is our opinion that the implementation of experiments addressing the shortcomings of the manuscript will take more than the allowable time granted for a revised manuscript. However, we would consider a revised paper if it fully addresses the reviewers' comments.

Reviewer #1:

In their paper "E3 Ubiquitin ligase Bre1 couples sister chromatid cohesion establishment to DNA replication in *Saccharomyces cerevisiae*", Zhang and colleagues investigate the underlying cause as to why yeast cells lacking the E3 ligase Bre1 and to a lesser degree its interacting partner Lge1 exhibit chromosome instability, as assayed by loss of a minichromosome fragment. The authors have performed experiments that suggest Bre1 is required to establish sister chromatid cohesion but not to maintain it. The mechanism as to how Bre1 is involved is not entirely resolved, although the authors test if Bre1 is required for localization/loading of a number of candidate factors.

Overall, I have a number of concerns regarding the author's experimental design and interpretation, and I am not convinced they have may a significant advance in our understanding of chromatin, cohesion or chromosome segregation as advertised in both the title and Abstract of the manuscript.

1) The authors main claim that Bre1 catalytic activity, and H2B ubiquitination, is required for cohesion establishment but this is not tested. In Figure 1, the authors use a cohesion maintenance assay. Thus, the only conclusion that can be drawn regarding catalytic activity and H2B based on the data shown is that they affect cohesion at some step. If the authors would like to implicate H2B and Bre1 catalytic activity in establishment, they need to test these alleles in their establishment assays used in Figure 1.

2) To determine how Bre1 is involved in cohesion establishment, the authors use chromosome spreads to localize a variety of replication and cohesion factors. They also perform ChIP on several finding decreased binding in *bre1∆* in certain cell cycle stages. Based on their images, it seems like there are two types of defects: reduced levels of protein and no staining. The authors only consider the second category but both could easily lead to defects. Additionally it is possible that loss of Bre1 affects the expression/levels of these proteins or nature of the HU arrest.

3) The presumption from these experiments is that the activity from Bre1 through H2BUbquitnation facilitates localization of Mcm10, Ctf4, Ctf18 and Eco10 to origins in G1 and S phases. It seems that the experiments needed to make this conclusion (and exclude other possible Bre1 targets) are to show alongside *bre1∆* that the same phenotypes are observed in *bre1∆*RING and in H2Bub mutant, or experiments that bypass the requirement for Bre1. Without these (and possibly even with them), it is unclear if the role of Bre1 is direct or indirect.

4) The authors use auxin to degrade Bre1 and claim to achieve ~90% reduction in activity. Their data suggests that degradation may be cell cycle dependent, an interesting finding the authors do not pursue. This issue complicates their experiments but because the authors never show how rapidly Bre1 recovers following auxin removal (which may also be cell cycle dependent), the experiments in Figure 2 rely on two weak assumptions: a) the premise that Bre1 is removed in this growth protocol with kinetics similar to the simple cell cycle arrests and b) it rapidly recovers as soon as auxin is removed. I would suggest westerns to accompany each time course at a minimum but would prefer microscopic data showing Bre1 levels remaining (using immunofluorescence) in cells with and without cohesion. This would allow the authors to strongly correlate premature separation with reduced Bre1.

5) Upon reflection, I wonder if the auxin allele is at all necessary in terms of the main points the authors address in the paper. They could in theory, use this allele to rule out Bre1 affects in replication/DNA damage, etc., but have only included it as a tool to study cohesion. In Figure 1, the authors convincingly showed the Bre1 played a role in the establishment but not maintenance of cohesion.

Reviewer #2:

In this manuscript, the authors explore possible mechanisms to account for the observation that mutations in the Bre1 ubiquitin ligase, and its interacting partner, Lge1, lead to a numerical CIN phenotype characterized by whole chromosome losses and gains. They focused on the relationship of Bre1 to sister chromatid cohesion, specifically cohesin complex proteins, and factors that load and activate cohesion proteins on chromatin. Using an established assay to measure the fidelity of chromosome transmission they first established that deletion of BRE1 or LGE1 resulted in an increase in chromosome loss. They then showed that this phenotype was related to a cohesion defect in these strains, as well as in an htb-K123R mutant, which cannot be monoubiquitinated by Bre1. To determine when in the cell cycle Bre1's role was important for sister chromatid cohesion, they employed an auxin-inducible degron system to control the expression of Bre1 in G1, S, and G2/M phases, and correlated expression with cohesion. They concluded that Bre1 plays a prominent role in cohesion in G1 and S phases. They then investigated if this phenotype was due to a role for Bre1 in the loading of cohesin in G1 phase or in the establishment of sister chromatid cohesion in S phase using chromatin spreads and ChIP of cohesion factors to replication origins in a *bre1∆* mutant. They conclude that cohesin loading occurs in the absence of Bre1, but that Bre1 is required for recruitment of several cohesion establishment factors to replication origins in S phase, and for the acetylation of the Smc3. Finally, using similar approaches, they found that Bre1 in part facilitated the origin recruitment of the replication initiation and elongation factor, Mcm10, which plays a role in both the recruitment of cohesion establishment factors and the activation/elongation of the replicative helicase.

There are several interesting observations that arise from this study. The genetic data showing that Bre1 plays an important role in suppressing chromosome loss through its function in the establishment of sister chromatid cohesion appear generally solid. This is new information that expands the cellular roles of this ubiquitin ligase in the maintenance of genome stability. The data additionally provide some support for the model that the establishment of sister chromatid cohesion by Bre1 could be coupled its role in DNA replication. There are, however, several points that significantly diminish enthusiasm for the study, as outlined below:

1) The authors posit that the effects of Bre1 on sister chromatid cohesion are likely to be mediated through its monoubiquitination of H2B, yet they only provided one piece of phenotypic evidence with an htb-K123R mutant (Figure 1) to support this statement. Bre1 could have other substrates, and in fact it has been reported to physically interact with the Smc1-3 cohesin proteins in a 2-hybrid assay. The authors showed that Smc3 association with chromatin was diminished in a *bre1∆* mutant and that Smc3 was also not efficiently acetylated and activated. Thus, one model that should be considered is that Smc3 or other Smc proteins (or other cohesion complex factors) could be modified (i.e. ubiquitinated) by their direct association with Bre1. Along the same lines, Mcm10 has been reported to be di-ubiquitinated in late G1-S, and this is required for its interaction with PCNA. Is this Bre1-dependent; does Bre1 interact with Mcm10?

2) A more serious concern is with the authors' experiments using a degron mutant of Bre1 to identify when in the cell cycle Bre1 is required for cohesion (Figure 2). First, they epitope tagged Bre1 on its C-terminus, which is known to eliminate its ligase activity. More importantly, Brian Strahl's lab reported in 2014 that mutations in Bre1 that affect its enzymatic activity significantly decrease its stability. He reported that this effect also extends to an htb-K123R mutation! This raises the question of whether the experiments in Figure 2 accurately reflect when Bre1 functions in the cell cycle to mediate cohesion.

3) The kinetics of recruitment of cohesion and replication factors to origins need to be followed, not just cells released into HU for 3 hours; e.g., release from α factor, nocodozole in WT and *bre1∆* cells. The association with these factors at sites away from the origin should also be incorporated to examine the role of Bre1 in RF progression.

[Editors’ note: what now follows is the decision letter after the authors submitted for further consideration.]

Thank you for submitting your article "E3 ubiquitin ligase Bre1 couples sister chromatid cohesion establishment to DNA replication in *Saccharomyces cerevisiae*" for consideration by *eLife*. Your manuscript has now been reviewed by two experts in the field and a member of our Board of Reviewing Editors. I am delighted to report that the reviewers and the BRE member are generally positive and have requested several minor revisions provided below:

1) Does a rad6 mutant exhibit a cohesion defect? This experiment was conspicuously absent.

2) Results section. "The distance between the two sister chromatids can be determined by visualizing the GFP signals during G2/M phase". This is not strictly true as the distance between sisters is not measured using the 1-spot/2 spot assay--rather the resolution of two spots indicates a loss of tight pairing between budding yeast chromosomes.

3) Also in the Results section. "Ctf4, Ctf18 and Eco1 associate with chromatin during all stages of the cell cycle". This statement is confusing as limiting Eco1 due to a cell cycle degradation in G2/M phase restrains de novo cohesion establishment (Lyons and Morgan, 2011).

4) The authors propose that "Bre1-mediated H2Bub1 marks replication origins to signal and couple DNA replication and cohesion establishment processes." (Section header) This is a strong statement that in my view is not justified by the data. That cohesion establishment occurs concomitantly with S phase in unchallenged cells has long been known (Skibbens et al., 1999; Tóth et al., 1999), but I'm not certain that the work presented in the manuscript using HU arrested cells (likely with collapsed replication forks) fundamentally extends this initial observation to the point of establishing the coupling between the two processes. Eco1 can promote cohesion establishment independently of DNA replication, hence the evidence for "coupling" of establishment with replication remains indirect. Until Smc3-Ac antibodies are available that can ChIP, it remains unclear if cohesion is in fact established during replication or considerably post-fork passage, for instance while the two sisters are still close together. The statement should thus be softened.

I have also included the reviewers' detailed comments at the end of the letter and would be interested to reading your thoughts on the comments of the reviewers.

Reviewer #1:

In this revision, the authors provide evidence that the E3 ligase, Bre1, plays a role in coupling DNA replication to the establishment of sister chromatid cohesion. The authors addressed the basis of the known chromosome instability phenotype of bre1 mutants. They report that Bre1, and H2B ubiquitination, which is controlled by Bre1, promote chromosome segregation and sister chromatid cohesion during the G1 and S phases of the cell cycle. This role is executed through the recruitment of cohesion establishment factors to chromatin and the acetylation of the cohesion factor, Smc3. Finally, they report that Bre1/H2Bub1 are required for the recruitment of replication factors that help to localize factors required for cohesion establishment. They additional found a novel role for Bre1/H2Bub1 in the recruitment of the replication initiation factor, Mcm10, to chromatin. The interaction of Mcm10 and replication factors with cohesion establishment factors supports a model in which the presence of Bre1 and H2Bub1 at origins signals the recruitment of both replication factors and cohesion establishment factors to chromatin, thereby coupling replication and sister chromatin cohesion.

In general, the data support the authors' main conclusions, which are based on a nice mix of genetic, molecular, and cell biological approaches. The results uncover a new function for Bre1 and H2Bub1 in replication and chromosome segregation, thereby extending the role of this histone modification in the cell cycle.

Reviewer #2:

The manuscript by Zheng et al. entitled "E3 ubiquitin ligase Bre1 couples sister chromatid cohesion establishment to DNA replication in *Saccharomyces cerevisiae*" demonstrates that H2B monoubiquitylation (at K123) promotes the S-phase establishment but not G2/M phase maintenance of sister chromatid cohesion. The authors show that global Smc3-Ac levels are decreased in bre1 mutants, consistent with this interpretation. While it was previously known that bre1 mutants exhibited a whole chromosome instability phenotype and defects in DNA replication/repair (Rizzardi et al., 2012; Trujillo and Osley, 2012), the authors here provide a plausible mechanism to explain CIN (although it should be noted that the data cannot rule out that defects in the completion of DNA replication would cause CIN via non-disjunction). Overall, this is a well-executed and interesting study that provides a reasonable though perhaps not wholly surprising mechanism for CIN in the absence of H2Bub1. To date, monoubiquitylation of H2B has been inferred to be required for fork stability in HU, however the observation of cohesion defects in unperturbed cycling cells in the absence of bre1 is an important step forward. The modest decrease in Eco1 recruitment to early ARSs (although much more dramatic in G1 arrested cells) is consistent with the decrease in global Smc3-acetylation that the authors observe (Figure 3), although the link between fork progression per se and cohesion establishment remains indirect.

Experimentally, my primary concern with this work is that HU arrests are used for ChIP and cell cycle-release experiments: as bre1 cells are known to exhibit HU sensitivity (death in HU), it is unclear how well these experiments reflect the mechanisms that underlie a failure to establish cohesion in cycling cells versus the loss of association of factors with collapsed forks. ChIP of replication factors over time post-G1 release into HU--pre-fork collapse as per Trujillo and Osley, 2012--would provide a more in depth view of replication factor dynamics and in the Trujillo work, appears to show greater sensitivity in detecting transient changes in replication factor association.

Another weakness of the manuscript is in clearly identifying mechanistic/causal relationships between H2Bub1 and cohesion establishment. The authors' model in Figure 6 implies that everything downstream of Bre1 will lead to a decrease in Eco1 recruitment, but Trujillo and Osley (2012) showed that PCNA levels at ARS305 are unaffected in the absence of H2B Ub (esp. in G1), and this is difficult to reconcile with the presented model. Moreover, the authors' most compelling data is that Bre1 likely confers its role in G1 phase, prior to the establishment reaction per se as sister chromatids have not yet been generated. This is an interesting observation, and could suggest that the establishment of cohesion involves not only the loading of cohesins, but also modulating some aspect of their function prior to cohesion establishment (the most obvious would be limiting turnover on chromatin--and this would fit with the authors' finding that wpl1∆ completely suppresses the cohesion defect in *bre1∆* cells, at least at the G1+60 minute time point--see specific comments).

Specific comments:

Results section: If 20% of *bre1∆* cells exhibit 2 spots (paragraph two), why then do only 9% of G1 cells exhibit 2 or no spots? Shouldn't this number be closer to 40% as each large budded cell in G2/M gives rise to 2 daughters--one with 2 spots and one with none? Are these cells failing to divide, possibly owing to tangled/unreplicated/damaged chromosomes?

Also in the Results section. "Taken together, these findings demonstrated that Bre1's role in SCC is most prominent in G1 phase". The statement is a bit misleading as there is no sister chromatid in G1 phase and thus there can be no direct role in SCC at this time. I agree with the authors' interpretation that events prior to the onset of replication are clearly playing a role in cohesion establishment, however I note that the authors' model should be expanded to consider how a "pre-establishment" function by H2Bub1 in G1 could be incorporated into the model (refer to Figure 6).

"Bre1 regulates the transcription of genes involved in the G1-S transition […] We constructed strains expressing Scc1 or Smc3 HA tagged and examined their chromatin enrichment in G1, S and G2/M" This is not the right experiment to determine if Bre1 regulates transcription of cohesion factors, particularly as chromatin spreads are notoriously non-quantitative. If Bre1 is involved in transcription of cohesin genes, then monitoring mRNA levels is a much better experiment and allows the authors to survey a broad collection of genes. Alternatively, protein levels ALL cohesin subunits and cohesion regulatory factors (including Rad61, Pds5, Scc2, Scc4, RFC components etc.) by quantitative Western blot would be necessary.

"*wpl1∆* partially rescues *bre1∆* cohesion defect at 60-75 min after G1 release […] This suggests that stabilizing cohesin alone on chromatin does not fully rescue cohesion defect in bre1 cells, as is the case for ctf4". This experiment was unclear--from the FACS data, the chromosomes are fully replicated by 60' and by 75-90' would be expected to progress past the metaphase to anaphase transition, leading to cohesion dissolution and a high% of cells with 2 spots. How was this experiment done--was nocodazole added to prevent cohesion dissolution, or are these cycling cells as indicated in the figure legend? Is Scc1 cleaved? If the cells are in fact released from α factor, then the 60' time point is the crucial one that shows full suppression of the PDS phenotype, leading one to postulate that H2Bub1 could play a role in decreasing cohesin turnover on chromatin prior to Smc3 acetylation, thus promoting cohesion establishment. This could be investigated by measuring Scc1 turnover on chromosomes +/- H2Bub in an SCC2-AID mutant to prevent cohesin re-loading.

"Yet, H2Bub1 is dispensable for the most upstream origin recognition complex component at origins (Trujillo and Osley, 2012)". In this same paper, Trujillo also showed that PCNA recruitment is not affected under these conditions--this does not fit with the authors’ model in Figure 6 and should be addressed experimentally or the model redesigned to take the data into account.

[Editors' note: further revisions were requested prior to acceptance, as described below.]

Thank you for resubmitting your work entitled "E3 ubiquitin ligase Bre1 couples sister chromatid cohesion establishment to DNA replication in *Saccharomyces cerevisiae*" for further consideration at *eLife*. Your revised article has been favorably evaluated by Kevin Struhl (Senior editor), Ali Shilatifard (Reviewing Editor), and one of the reviewers. The manuscript has been improved but there are some remaining issues that need to be addressed (as outlined below) before acceptance.

Please address the following comments/suggestions:

1) Present data on the effect of a *bre1∆* mutant on the expression of a larger cohort of cohesion genes. The reviewer and BRE request that you include RT-PCR data on the expression of cohesion related genes as previously requested.

2) Please present western blot data for Bre1 protein levels in Bre1's catalytic activity mutants as well as in the bre1-RING deletion mutant.

---

## [Author Response]

[Editors’ note: the author responses to the first round of peer review follow.]

Reviewer #1:[…] Overall, I have a number of concerns regarding the author's experimental design and interpretation, and I am not convinced they have may a significant advance in our understanding of chromatin, cohesion or chromosome segregation as advertised in both the title and Abstract of the manuscript.1) The authors main claim that Bre1 catalytic activity, and H2B ubiquitination, is required for cohesion establishment but this is not tested. In Figure 1, the authors use a cohesion maintenance assay. Thus, the only conclusion that can be drawn regarding catalytic activity and H2B based on the data shown is that they affect cohesion at some step. If the authors would like to implicate H2B and Bre1 catalytic activity in establishment, they need to test these alleles in their establishment assays used in Figure 1.

We thank the reviewer for pointing out the difference in the assays. We have toned down our conclusion for Figure 1, saying that the catalytic activity of Bre1 and H2B are important for cohesion, either at establishment or maintenance. More importantly, we have complementarily performed G1 arrest and release time course cohesion assay for wild-type, *bre1Δ, bre1-RINGΔ* and *htb1-K123R htb2Δ* mutants, as shown in Figure 1, as in Hanna et al., 2001, and in Figure 1 but with more time points. Wild-type maintains low frequencies of sister chromatid separation from 0-90 min after G1 release, whereas *bre1Δ, bre1-RINGΔ* and *htb1-K123R htb2Δ* mutants show similar progressive increase in premature sister chromatid separation starting from 30 min after G1 release, suggesting that these mutants have defects since cohesion establishment, instead of having defects only in cohesion maintenance. Based on previous literatures such as Hanna et al., 2001, we noted that in G1 arrest and release cohesion time course assay, the premature sister chromatid separation frequency in S phase is significant, but lower than that in G2/M phase even for well-studied cohesion establishment factors, such as *ctf18Δ*.

2) To determine how Bre1 is involved in cohesion establishment, the authors use chromosome spreads to localize a variety of replication and cohesion factors. They also perform ChIP on several finding decreased binding in bre1∆ in certain cell cycle stages. Based on their images, it seems like there are two types of defects: reduced levels of protein and no staining. The authors only consider the second category but both could easily lead to defects. Additionally it is possible that loss of Bre1 affects the expression/levels of these proteins or nature of the HU arrest.

We have checked that *BRE1* deletion does not reduce the total protein levels of Myc- or HA-tagged replication or cohesion establishment factors (Polα, Psf2 and Mcm10, Ctf4, Ctf18 and Eco1), as shown by western blot analyses in Figure 3—figure supplement 2 and Figure 4—figure supplement 1. For chromatin spreads, our quantification is based on a binary assay for each DAPI chromatin mass, by analyzing whether the DAPI-stained chromatin mass is associated with replication or cohesion factor signal spot, or not. The spot sizes for the replication or cohesion factors are similar to those of the DAPI masses, and the intensity of the factors are uniform in the category in which DAPI-stained chromatin mass is associated with replication or cohesion factor. Then, we calculated the percentage of chromatin masses associated with replication or cohesion factors. We have relabeled the Y-axis as “% chromatin masses associated with the specific protein” for clarification. The loss of Bre1 decreases the percentage of chromatin-associated replication factors and cohesion establishment factors in G1 or S phase by chromatin spreads (Figure 3 and Figure 4). The chromatin spread assay prompted us to further investigate by ChIP-qPCR whether Bre1 affects the occupancy of replication factors and cohesion establishment factors at early origins and early origin-flanking region in HU-arrested S phase (Figure 3 and Figure 4) or G1 phase (Figure 3 and Figure 4).

For the experiments involving HU arrest, *bre1Δ* mutant indeed showed a slightly delayed G1-S transition (based on bud formation, data not shown and (Jorgensen et al., 2002; Ni and Snyder, 2001)). It took slightly more time for HU-induced S phase arrest in *bre1Δ* mutant. However, after arresting cells in HU for 3 hours, about 95% of both WT and *bre1Δ* mutant cells showed budded cells and similar FACs profiles (Figure 3—figure supplement 2 and Figure 4—figure supplement 1). These comparable cells were subjected to the chromatin spread assay. This excluded the possibility that loss of Bre1 affects the nature of HU arrest and the chromatin spread assay.

3) The presumption from these experiments is that the activity from Bre1 through H2BUbquitnation facilitates localization of Mcm10, Ctf4, Ctf18 and Eco10 to origins in G1 and S phases. It seems that the experiments needed to make this conclusion (and exclude other possible Bre1 targets) are to show alongside bre1∆ that the same phenotypes are observed in bre1∆RING and in H2Bub mutant, or experiments that bypass the requirement for Bre1. Without these (and possibly even with them), it is unclear if the role of Bre1 is direct or indirect.

To test if Swd2, the only other known ubiquitination target of Bre1 (Vitaliano-Prunier et al., 2008), functions in cohesion, we have performed cohesion assay on *swd2Δ* after nocodazole arrest, and found that *swd2Δ* does not lead to premature sister chromatid separation (Figure 1).

In addition, we have checked the early origin occupancy of replication factor Mcm10 and cohesion establishment factor Ctf4 in *bre1-RINGΔ or htb1-K123R htb2∆* mutants in HU-arrested cells by ChIP, and found that these mutants have also reduced the occupancy of Mcm10 and Ctf4 at two early origins (Figure 5), as in *bre1Δ*, suggesting that the effect of Bre1 on cohesion establishment through replication can be attributed to the catalytic RING domain and substrate H2B monoubiquitination modification.

4) The authors use auxin to degrade Bre1 and claim to achieve ~90% reduction in activity. Their data suggests that degradation may be cell cycle dependent, an interesting finding the authors do not pursue. This issue complicates their experiments but because the authors never show how rapidly Bre1 recovers following auxin removal (which may also be cell cycle dependent), the experiments in Figure 2 rely on two weak assumptions: a) the premise that Bre1 is removed in this growth protocol with kinetics similar to the simple cell cycle arrests and b) it rapidly recovers as soon as auxin is removed. I would suggest westerns to accompany each time course at a minimum but would prefer microscopic data showing Bre1 levels remaining (using immunofluorescence) in cells with and without cohesion. This would allow the authors to strongly correlate premature separation with reduced Bre1.

We have checked the Bre1-AID*-9Myc degradation and recovery time course for G1, G1+S or S degradation schemes in Figure 2, which involve the recovery step and show cohesion defects in G2/M (Figure 2) by western blot analyses (Figure 2). Bre1-AID*-9Myc can be recovered to > 10% of the original level within 120 min after removal of auxin. We collected G2/M cells after 240 min in nocodazole for assessing cohesion, when most Bre1 protein has recovered from degradation. This important control has allowed this assay to be used to estimate Bre1’s functional time for cohesion during the cell cycle.

5) Upon reflection, I wonder if the auxin allele is at all necessary in terms of the main points the authors address in the paper. They could in theory, use this allele to rule out Bre1 affects in replication/DNA damage, etc., but have only included it as a tool to study cohesion. In Figure 1, the authors convincingly showed the Bre1 played a role in the establishment but not maintenance of cohesion.

The auxin-degron assay was originally used as a screening assay to conditionally control the degradation of Bre1 during specific cell cycle stage, thereby distinguishing the stage at which Bre1 is important for cohesion. The results have guided us towards investigating cohesin loading in G1 phase and cohesion establishment in S phase. As pointed out by reviewer #1 in major point 1, the nocodazole arrest cohesion assay only tells whether Bre1 functions in cohesion in G2/M, while the G1 arrest and release cohesion assay suggests whether the mutant has defect starting in S phase or only in G2/M. Now with the G1 arrest and release cohesion assay on all mutants (Figure 1) and solid data that Bre1 functions in G1 and S for replication/cohesion factor association to chromatin (Figure 3 and Figure 4), we agree that the results of the auxin-degron assay are complementary and supporting.

We agree with reviewer #1 that we could have used this degron allele to understand when is Bre1 required for replication and DNA damage. Here, we chose to focus on one novel function, cohesion. It turns out the cohesion function is tightly linked with the replication function.

Reviewer #2:[…] 1) The authors posit that the effects of Bre1 on sister chromatid cohesion are likely to be mediated through its monoubiquitination of H2B, yet they only provided one piece of phenotypic evidence with an htb-K123R mutant (Figure 1) to support this statement. Bre1 could have other substrates, and in fact it has been reported to physically interact with the Smc1-3 cohesin proteins in a 2-hybrid assay. The authors showed that Smc3 association with chromatin was diminished in a bre1∆ mutant and that Smc3 was also not efficiently acetylated and activated. Thus, one model that should be considered is that Smc3 or other Smc proteins (or other cohesion complex factors) could be modified (i.e. ubiquitinated) by their direct association with Bre1. Along the same lines, Mcm10 has been reported to be di-ubiquitinated in late G1-S, and this is required for its interaction with PCNA. Is this Bre1-dependent; does Bre1 interact with Mcm10?

As mentioned in response to reviewer #1 major point 1 and 3, we have performed G1 arrest and release time course cohesion assay for wild-type, *bre1Δ, bre1-RINGΔ* and *htb1-K123R htb2Δ* mutants, as shown in Figure 1. Wild-type maintains low frequency of sister chromatid separation from 0-90 min after G1 release, whereas *bre1Δ, bre1-RINGΔ* and *htb1-K123R htb2Δ* mutants show similar progressive increase in premature sister chromatid separation starting from 30 min after G1 release, suggesting that these mutants have defects since cohesion establishment, instead of having defects only in cohesion maintenance. In addition, we have tested if Swd2, the only other known ubiquitination target of Bre1 (Vitaliano-Prunier et al., 2008), functions in cohesion by the cohesion assay on *swd2Δ* after nocodazole arrest. We found that *swd2Δ* does not lead to premature sister chromatid separation (Figure 1).

As mentioned by reviewer #2 in the first paragraph, “cohesin loading occurs in the absence of Bre1”. Our chromatin spread assay showed that the association of cohesin subunit Smc3 with chromatin is not diminished in *bre1Δ* (Figure 3—figure supplement 1), but Bre1 is required for the chromatin association of cohesion establishment factors Ctf4, Ctf18 and Eco1 (Figure 3) and Smc3 acetylation (Figure 3). We have tested if cohesin subunits Smc3, or replication factor Mcm10 interacts with Bre1 by co-immunoprecipitation, and found that Bre1 does not interact with Smc3,but it interacts with Mcm10 weakly in our tested condition (Figure 3—figure supplement 1 and Figure 4—figure supplement 1).

Mcm10 has been reported to be di-ubiquitinated in late G1-S, and this is required for its interaction with PCNA (Das-Bradoo et al., 2006). Mcm10 is mono-ubiquitinated at two lysines, K85 and K372 (Zhang et al., 2016). DDB1-dependent E3 ligase, Cullin 4-based E3 ligase (CRL4), ubiquitinates Mcm10in vitro (Kaur et al., 2012). We have tested if Mcm10 ubiquitination level is affected in *bre1Δ* mutant by immunoprecipitating Mcm10 and analyzing by anti-ubiquitin antibody in Western blot (Figure 4—figure supplement 1). We found that Bre1 is dispensable the ubiquitination pattern of Mcm10 in budding yeast.

2) A more serious concern is with the authors' experiments using a degron mutant of Bre1 to identify when in the cell cycle Bre1 is required for cohesion (Figure 2). First, they epitope tagged Bre1 on its C-terminus, which is known to eliminate its ligase activity. More importantly, Brian Strahl's lab reported in 2014 that mutations in Bre1 that affect its enzymatic activity significantly decrease its stability. He reported that this effect also extends to an htb-K123R mutation! This raises the question of whether the experiments in Figure 2 accurately reflect when Bre1 functions in the cell cycle to mediate cohesion.

The C-terminus TAP tag of Bre1 affects Bre1’s interaction with Rad6 (104), but the C-terminus HA tag of Bre1, Rad6 and Lge1 genes does not affect the enzymatic activities of the corresponding proteins (Bonizec et al., 2014). To check the function of Bre1-AID*-9Myc, we checked the growth rate, cohesion defect, transcription targets’ levels and HU sensitivity. The growth rate and cohesion defect of Bre1-AID*-9Myc are comparable to those in wild-type (Figure 2), suggesting that AID*-9Myc tag does not affect growth and cohesion function. Importantly, in Figure 2, the Bre1-AID*-9Myc strain does not show cohesion defect under conditions with the addition of cell cycle arrest drugs, which is used as our baseline for comparison with specific degradation schemes.

Since Bre1 regulates the transcription of some cyclin genes, such as CLN2 and CLB5 (Zimmermann et al., 2011), we have checked their transcription levels in WT and Bre1-AID*-9Myc cells by RT-qPCR, and found that AID*-9Myc tag of Bre1 does not affect transcription of CLN2, but slightly reduces CLB5 transcription (Figure 2—figure supplement 1). We have tested the HU sensitivity of Bre1-AID*-9Myc together with WT, *bre1Δ* and *bre1-RINGΔ* strains (Figure 2—figure supplement 1), and unfortunately found that Bre1-AID*-9Myc has similar HU sensitivity as *bre1Δ* and *bre1-RINGΔ* strains, suggesting that the AID*-9Myc tag of Bre1 may disrupt some function of Bre1 in responding to replication stress. The wild-type-like phenotypes of Bre1-AID*-9Myc cells in growth and cohesion assays contrast with their minor defect in transcription and high HU sensitivity, suggesting that the AID*-9Myc tag may affect Bre1’s function in transcription and replication stress response more than that in cohesion, or that these assay sensitivities may be different.

Moreover, as suggested by reviewer #1 major point 4, we have checked when Bre1-AID*-9Myc is resynthesized after auxin is removed (Figure 2), to verify that the timing of Bre1 absence in these degradation schemes. Our baseline control in the cohesion assay (Figure 2) should allow us to conclude that the cohesion defect observed in Figure 2 is due to Bre1 degradation, but not only due to the AID*-9Myc tag. Now with the G1 arrest and release cohesion assay on all mutants (Figure 1) and solid data that Bre1 functions in G1 and S for replication/cohesion factor association to chromatin (Figure 3 and Figure 4), we agree that the results of the auxin-degron assay are complementary and supporting.

3) The kinetics of recruitment of cohesion and replication factors to origins need to be followed, not just cells released into HU for 3 hours; e.g., release from α factor, nocodozole in WT and bre1∆ cells. The association with these factors at sites away from the origin should also be incorporated to examine the role of Bre1 in RF progression.

The chromatin spread data for cohesion factors and replication factors showed that Bre1 does not affect the overall association of these factors with chromatin in G2/M phase, thus, we did not follow their enrichment in origin in G2/M phase by ChIP. However, since the association of Eco1 and Mcm10 with chromatin at G1 phase is reduced in *bre1Δ* by chromatin spread (Figure 3 and Figure 4), the enrichment of Eco1 and Mcm10 at early and late origins in α factor-arrested G1 phase in *bre1Δ* have been checked by ChIP. Consistently, Eco1 and Mcm10 enrichment at the two early origins (ARS305 and ARS306) are diminished, but those at a late origin (ARS501) are unaffected (Figure 3 and Figure 4).

For Eco1 and Mcm10, we have also checked their enrichment at sites 1.5 kb away from the early origin ARS305 in S phase, and found that their occupancies are also reduced.

[Editors' note: the author responses to the re-review follow.]

[…] 1) Does a rad6 mutant exhibit a cohesion defect? This experiment was conspicuously absent.

We have constructed *rad6Δ* and performed cohesion assay in nocodazole-arrested cells and G1-arrested and released cells, and found that *rad6Δ* exhibits similar cohesion defect as *bre1Δ* (Figure 5 and Figure 5—figure supplement 1), suggesting that E2 conjugating enzyme Rad6 works with E3 ubiquitin ligase Bre1 to monoubiquitinate H2B (Robzyk, Recht and Osley, 2000) and functions in cohesion.

2) Results section. "The distance between the two sister chromatids can be determined by visualizing the GFP signals during G2/M phase". This is not strictly true as the distance between sisters is not measured using the 1-spot/2 spot assay--rather the resolution of two spots indicates a loss of tight pairing between budding yeast chromosomes.

We have rephrased the sentence to “The separation of the two sister chromatids can be visualized by the GFP signals during G2/M phase.”

3) Also in the Results section. "Ctf4, Ctf18 and Eco1 associate with chromatin during all stages of the cell cycle". This statement is confusing as limiting Eco1 due to a cell cycle degradation in G2/M phase restrains de novo cohesion establishment (Lyons and Morgan, 2011).

Hanna et al., 2001 showed that Ctf4 and Ctf18 associate with chromatin in α factor-, HU- and nocodazole-arrested cells by chromatin spread and fractionation (Figure 5 in Hanna et al., 2001). Tóth et al., 1999 showed that Eco1 is associated with chromatin in cycling cells by cell fractionation and chromosome spread (Figure 4 and data not shown in Tóth et al., 1999). We quantified our chromatin spread results in the 3 arrested states for Ctf4, Ctf18 and Eco1, and recapitulated the results for Ctf4 and Ctf18 in *WT* cells as in Hanna et al., 2011 and Eco1 as in Tóth et al., 1999.

On the other hand, Lyons and Morgan, 2011 monitored the protein and phosphorylation level of Eco1 to unravel the function of Eco1 phosphorylation in its degradation in G2/M phase. We suspect that we could not detect the degradation of Eco1 in G2/M phase because: 1) the degradation is not complete (Figure 3 in Lyons and Morgan 2011); and 2) the chromatin spread assay quantified the chromatin masses (100%) that are associated with or without Ctf4, Ctf18 and Eco1 spots, and thus may not be sensitive to detect the absolute change in the level of Eco1 at different states.

To avoid confusion, we have rephrased the sentence to “To elucidate whether Bre1 affects the recruitment of cohesion establishment factors Ctf4, Ctf18 and Eco1 to chromatin, chromatin spread was performed at different arrested cell cycle stages, and we found that Ctf4, Ctf18 and Eco1 associate with chromatin during all stages of the cell cycle in *WT* cells (Figure 3—figure supplement 2). This observation is consistent with previous studies (Tóth et al., 1999, Hanna et al., 2001), but our chromatin spread assay did not detect the degradation of Eco1 in G2/M phase as shown in a previous study (Lyons and Morgan, 2011).”

4) The authors propose that "Bre1-mediated H2Bub1 marks replication origins to signal and couple DNA replication and cohesion establishment processes." (Section header) This is a strong statement that in my view is not justified by the data. That cohesion establishment occurs concomitantly with S phase in unchallenged cells has long been known (Skibbens et al., 1999; Tóth et al., 1999), but I'm not certain that the work presented in the manuscript using HU arrested cells (likely with collapsed replication forks) fundamentally extends this initial observation to the point of establishing the coupling between the two processes. Eco1 can promote cohesion establishment independently of DNA replication, hence the evidence for "coupling" of establishment with replication remains indirect. Until Smc3-Ac antibodies are available that can ChIP, it remains unclear if cohesion is in fact established during replication or considerably post-fork passage, for instance while the two sisters are still close together. The statement should thus be softened.

To soften the statement and more accurately describe our findings, we have rephrased the Discussion section header to “Bre1-mediated H2Bub1 marks replication origins and recruits DNA replication factors in α-factor and HU arrest, facilitating cohesion establishment factors association and cohesion establishment in S phase”.

As responded below to reviewer #2’s second comment, we have included (in last revised manuscript) chromatin immunoprecipitation of Eco1 and Mcm10 at origins (Figure 3 and Figure 4, respectively) in α-factor-arrested cells, suggesting that Bre1’s effect on Mcm10 and Eco1 does not only occur in collapsed forks, but also in G1 phase. This is consistent with *bre1* degron mutant results (Figure 2), suggesting that Bre1’s function in G1 phase is important for cohesion.

Reviewer #2:The manuscript by Zheng et al. entitled "E3 ubiquitin ligase Bre1 couples sister chromatid cohesion establishment to DNA replication in Saccharomyces cerevisiae" demonstrates that H2B monoubiquitylation (at K123) promotes the S-phase establishment but not G2/M phase maintenance of sister chromatid cohesion. The authors show that global Smc3-Ac levels are decreased in bre1 mutants, consistent with this interpretation. While it was previously known that bre1 mutants exhibited a whole chromosome instability phenotype and defects in DNA replication/repair (Rizzardi et al., 2012; Trujillo and Osley, 2012), the authors here provide a plausible mechanism to explain CIN (although it should be noted that the data cannot rule out that defects in the completion of DNA replication would cause CIN via non-disjunction). Overall, this is a well-executed and interesting study that provides a reasonable though perhaps not wholly surprising mechanism for CIN in the absence of H2Bub1. To date, monoubiquitylation of H2B has been inferred to be required for fork stability in HU, however the observation of cohesion defects in unperturbed cycling cells in the absence of bre1 is an important step forward. The modest decrease in Eco1 recruitment to early ARSs (although much more dramatic in G1 arrested cells) is consistent with the decrease in global Smc3-acetylation that the authors observe (Figure 3), although the link between fork progression per se and cohesion establishment remains indirect.

To reflect that Bre1’s function in replication, specifically at origins, has been implicated in minichromosome maintenance, we have edited the sentence to “While the structural CIN phenotype involving gross chromosomal rearrangements (GCR) observed in *bre1Δ* and *lge1Δ* can be explained by the known functions of H2Bub1 in DNA damage response and repair, the underlying cause of numerical CIN phenotype involving whole chromosome gains or losses in *bre1Δ* and *lge1Δ* is currently not clear, though Bre1’s function in replication origins has been implicated in minichromosome maintenance (Rizzardi et al., 2012).”

Experimentally, my primary concern with this work is that HU arrests are used for ChIP and cell cycle-release experiments: as bre1 cells are known to exhibit HU sensitivity (death in HU), it is unclear how well these experiments reflect the mechanisms that underlie a failure to establish cohesion in cycling cells versus the loss of association of factors with collapsed forks. ChIP of replication factors over time post-G1 release into HU--pre-fork collapse as per Trujillo and Osley, 2012--would provide a more in depth view of replication factor dynamics and in the Trujillo work, appears to show greater sensitivity in detecting transient changes in replication factor association.

We understand the reviewer’s concern and the suggestion, but this will involve repeating all the chromatin spread and chromatin immunoprecipitation experiments in Figure 3, Figure 4 and Figure 5 in time courses. Instead, in last revised manuscript, we have included chromatin immunoprecipitation of Eco1 and Mcm10 at origins (Figure 3 and Figure 4, respectively) in α-factor-arrested cells, suggesting that Bre1’s effect on Mcm10 and Eco1 does not only occur in collapsed forks, but also in G1 phase. This is consistent with *bre1* degron mutant results (Figure 2), suggesting that Bre1’s function in G1 phase is important for cohesion establishment in S phase, emphasizing the “pre-establishment function”, as discussed in the specific comment 2 below.

Another weakness of the manuscript is in clearly identifying mechanistic/causal relationships between H2Bub1 and cohesion establishment. The authors' model in Figure 6 implies that everything downstream of Bre1 will lead to a decrease in Eco1 recruitment, but Trujillo and Osley (2012) showed that PCNA levels at ARS305 are unaffected in the absence of H2B Ub (esp. in G1), and this is difficult to reconcile with the presented model. Moreover, the authors' most compelling data is that Bre1 likely confers its role in G1 phase, prior to the establishment reaction per se as sister chromatids have not yet been generated. This is an interesting observation, and could suggest that the establishment of cohesion involves not only the loading of cohesins, but also modulating some aspect of their function prior to cohesion establishment (the most obvious would be limiting turnover on chromatin--and this would fit with the authors' finding that wpl1∆ completely suppresses the cohesion defect in bre1∆ cells, at least at the G1+60 minute time point--see specific comments).

To avoid confusion, we added in Figure 6 figure legend that the effect of Bre1 on Mcm10, Ctf4, Ctf18 and Eco1’s recruitment to origins are partial (to indicate the incomplete reduction in chromatin spread assays), and removed the Mcm complex, PCNA and Polε in Figure 6 as we have not tested these components. We also modified the sentence in the Discussion to “The partially reduced level of chromatin-associated replication factors (Psf2 and Mcm10) in *BRE1* null mutant affects the localization of Ctf4, which in turn affect the localization of Ctf18, and thus Eco1, leading to reduced Smc3 acetylation and resulting in defective cohesion establishment. Surprisingly, PCNA’s association at early origins was unaffected in the absence of H2Bub1 (Trujillo and Osley 2012), and this could be because the partial reduction of Ctf18’s chromatin association is not severe enough, or that PCNA can associate to origins through a Ctf18-independent pathway.”

Specific comments:Results section: If 20% of bre1∆ cells exhibit 2 spots (paragraph two), why then do only 9% of G1 cells exhibit 2 or no spots? Shouldn't this number be closer to 40% as each large budded cell in G2/M gives rise to 2 daughters--one with 2 spots and one with none? Are these cells failing to divide, possibly owing to tangled/unreplicated/damaged chromosomes?

We speculate that premature cohesion defect in G2/M phase is always stronger than chromosome missegregation defect in G1 phase because there is a chance that the premature sister chromatids may randomly segregate correctly (say ~50% chance among the 20% premature separation cases, resulting in only ~10% missegregated cells), though the mechanism of this random segregation is not clear.

Also in the Results section. "Taken together, these findings demonstrated that Bre1's role in SCC is most prominent in G1 phase". The statement is a bit misleading as there is no sister chromatid in G1 phase and thus there can be no direct role in SCC at this time. I agree with the authors' interpretation that events prior to the onset of replication are clearly playing a role in cohesion establishment, however I note that the authors' model should be expanded to consider how a "pre-establishment" function by H2Bub1 in G1 could be incorporated into the model (refer to Figure 6).

In the sentence “Taken together, these findings demonstrated that Bre1’s role in sister chromatid cohesion is most prominent in G1 phase, but also in S phase, consistent with the timing of cohesin loading in G1 phase or cohesion establishment in S phase, but Bre1 is not required in G2/M phase for cohesion maintenance.” In the second part of the sentence, we tried to to explain our initial hypothesis by matching the timing of Bre1’s function in G1 and S phase to known events in the cohesion cycle. We hope the readers will get the idea that G1 phase is before cohesion establishment in S phase. In addition, we have added the recruitment of Mcm10 and Ctf4 in G1 phase in Figure 6 to indicate this ‘pre-establishment” function.

"Bre1 regulates the transcription of genes involved in the G1-S transition […] We constructed strains expressing Scc1 or Smc3 HA tagged and examined their chromatin enrichment in G1, S and G2/M" This is not the right experiment to determine if Bre1 regulates transcription of cohesion factors, particularly as chromatin spreads are notoriously non-quantitative. If Bre1 is involved in transcription of cohesin genes, then monitoring mRNA levels is a much better experiment and allows the authors to survey a broad collection of genes. Alternatively, protein levels ALL cohesin subunits and cohesion regulatory factors (including Rad61, Pds5, Scc2, Scc4, RFC components etc.) by quantitative Western blot would be necessary.

We are not suggesting that Bre1 would affect the transcription of cohesin components. Indeed, we verified that cohesion components are not among the G1-S transition genes regulated by Bre1 in Zimmermann et al., 2011. Consistently, our Western blot results in Figure 3—figure supplement 1 and B show that the protein levels of Scc1 and Smc3 are not affected.

To clarify our hypothesis, we rephrased this sentence to “Cohesin associates with chromatin in late G1, then accumulates at regions of convergent transcription (Lengronne et al., 2004). As Bre1 regulates the transcription of genes involved in G1-S transition (Zimmermann et al., 2011), the change in transcription pattern may affect the binding of cohesin on chromatin.”

"wpl1∆ partially rescues bre1∆ cohesion defect at 60-75 min after G1 release […] This suggests that stabilizing cohesin alone on chromatin does not fully rescue cohesion defect in bre1 cells, as is the case for ctf4". This experiment was unclear--from the FACS data, the chromosomes are fully replicated by 60' and by 75-90' would be expected to progress past the metaphase to anaphase transition, leading to cohesion dissolution and a high% of cells with 2 spots. How was this experiment done--was nocodazole added to prevent cohesion dissolution, or are these cycling cells as indicated in the figure legend? Is Scc1 cleaved? If the cells are in fact released from α factor, then the 60' time point is the crucial one that shows full suppression of the PDS phenotype, leading one to postulate that H2Bub1 could play a role in decreasing cohesin turnover on chromatin prior to Smc3 acetylation, thus promoting cohesion establishment. This could be investigated by measuring Scc1 turnover on chromosomes +/- H2Bub in an SCC2-AID mutant to prevent cohesin re-loading.

We did not add nocodazole to arrest cells, so these are cycling cells. Based on Figure 1 FACS profiles, most cells are still in G2/M in 60-75 minutes, and only a small portion of cells start to go to G1 in 90 minutes, thus only ~5% *WT* cells have separated GFP spots (Figure 5 and Figure 3—figure supplement 2). We added this possibility to the Discussion; “Surprisingly, *wpl1Δ* partially rescues *bre1Δ*’s cohesion defect at 60-75 min after G1 release, suggesting Bre1 could play a role in reducing cohesion turnover on chromatin, counteracting Wpl1 (Figure 3—figure supplement 2). Alternatively, it may reflect that the cell cycle progression of *wpl1Δ bre1Δ* is slightly delayed (Figure 3—figure supplement 2). However, *wpl1Δ* has 9.2% cohesion defect at 90 min after G1 release, whereas *wpl1Δ bre1Δ* has 18.8%, similar to that in *bre1Δ* alone. This suggests that stabilizing cohesin alone on chromatin does not fully rescue cohesion defect in *bre1Δ*, as in the case for *ctf4Δ* (Borges et al., 2013).”

"Yet, H2Bub1 is dispensable for the most upstream origin recognition complex component at origins (Trujillo and Osley, 2012)". In this same paper, Trujillo also showed that PCNA recruitment is not affected under these conditions--this does not fit with the authors’ model in Figure 6 and should be addressed experimentally or the model redesigned to take the data into account.

See above responses to reviewer #2’s general comments and the editions in Figure 6.

[Editors' note: further revisions were requested prior to acceptance, as described below.]

[…] 1) Present data on the effect of a bre1∆ mutant on the expression of a larger cohort of cohesion genes. The reviewer and BRE request that you include RT-PCR data on the expression of cohesion related genes as previously requested.

We have monitored the mRNA levels of genes that we have tested by chromatin spreads and ChIP-qPCR in our manuscript, including the cohesin component genes, *SCC1* and *SMC3*, and cohesion establishment genes, *CTF4, CTF18* and *ECO1*, in wild-type and *bre1∆* asynchronous log phase cells by RT-qPCR as suggested. We found that the mRNA expression levels of *SCC1, SMC3, CTF4, CTF18* and *ECO1* were not affected in the absence of *BRE1* (Figure 3—figure supplement 1).

2) Please present western blot data for Bre1 protein levels in Bre1's catalytic activity mutants as well as in the bre1-RING deletion mutant.

We have tagged endogenous full-length Bre1 (in wild-type and *htb1-K123R htb2Δ* mutant) and *bre1-RINGΔ* at C-terminus with 3HA, as in (Bonizec et al., 2014), and checked the endogenous Bre1 protein levels in asynchronous log phase cells of these strains. Consistent with (Wozniak and Strahl 2014), we found that *bre1-RINGΔ-3HA* mutant Bre1 protein level was reduced when compared with wild-type (Bre1-3HA) (Figure 5—figure supplement 2), suggesting that Bre1 RING domain affects its catalytic activity as well as Bre1 stability. However, Bre1 level in *htb1-K123R htb2Δ* was comparable to that in WT (Figure 5—figure supplement 2). This supports our conclusion that the effect of Bre1 on H2B monubiquitination can account for the cohesion defects in *bre1Δ* or *htb1-K123R* mutants.